# Resveratrol for the Management of Human Health: How Far Have We Come? A Systematic Review of Resveratrol Clinical Trials to Highlight Gaps and Opportunities

**DOI:** 10.3390/ijms25020747

**Published:** 2024-01-06

**Authors:** Karen Brown, Despoina Theofanous, Robert G. Britton, Grandezza Aburido, Coral Pepper, Shanthi Sri Undru, Lynne Howells

**Affiliations:** 1Leicester Cancer Research Centre, University of Leicester, Robert Kilpatrick Clinical Sciences Building, Leicester Royal Infirmary, Leicester LE2 7LX, UK; dt210@leicester.ac.uk (D.T.); rgb6@le.ac.uk (R.G.B.); grandezza.aburido@yahoo.co.uk (G.A.); ssu4@leicester.ac.uk (S.S.U.); lh28@leicester.ac.uk (L.H.); 2Odames Library, Victoria Building, Leicester Royal Infirmary, Leicester LE1 5WW, UK

**Keywords:** resveratrol, clinical trials, systematic review, nutraceuticals

## Abstract

Resveratrol has long been proposed as being beneficial to human health across multiple morbidities, yet there is currently no conclusive clinical evidence to advocate its recommendation in any healthcare setting. A large cohort with high-quality clinical data and clearly defined biomarkers or endpoints are required to draw meaningful conclusions. This systematic review compiles every clinical trial conducted using a defined dose of resveratrol in a purified form across multiple morbidities to highlight the current ‘state-of-play’ and knowledge gaps, informing future trial designs to facilitate the realisation of resveratrol’s potential benefits to human health. Over the last 20 years, there have been almost 200 studies evaluating resveratrol across at least 24 indications, including cancer, menopause symptoms, diabetes, metabolic syndrome, and cardiovascular disease. There are currently no consensus treatment regimens for any given condition or endpoint, beyond the fact that resveratrol is generally well-tolerated at a dose of up to 1 g/day. Additionally, resveratrol consistently reduces inflammatory markers and improves aspects of a dysregulated metabolism. In conclusion, over the last 20 years, the increasing weight of clinical evidence suggests resveratrol can benefit human health, but more large, high-quality clinical trials are required to transition this intriguing compound from health food shops to the clinic.

## 1. Introduction

There have been over 13,000 publications featuring resveratrol (RSV) in the title since the seminal paper by Jang et al. that demonstrated its potential cancer preventive effects in 1997 [1]. Resveratrol has been pursued for a vast array of clinical indications and health benefits, but the majority of research in this context has been at the preclinical level in simple cellular and animal models. In the laboratory, resveratrol has been reported to exert cardioprotective, neuroprotective, antitumour, antidiabetic, antibacterial, and anti-ageing effects; common to some of these effects is an ability to modulate glucose metabolism, oxidative stress, cell death, and inflammation, amongst numerous potential mechanisms of action. However, to ascertain the potential value of resveratrol as a supplement for humans, clinical studies, and ultimately randomised controlled trials (RCTs), are essential to provide critical evidence that may justify its recommendation.

Conducting meaningful RCTs with dietary-derived compounds is fraught with challenges, particularly around the lack of financial incentives and long-term returns for nutraceutical companies that would justify funding large expensive trials, akin to the model of drug development followed by the pharmaceutical industry. Additional barriers arise from lack of patentability, manufacturing difficulties, and differences in regulatory requirements, as well as the type of health claims that can be made for nutraceuticals versus investigational medicinal products, product consistency, and quality control. 

Trials using dietary-derived compounds typically have to be grant-funded through charitable or government organisations and, as such, they are often limited to small-scale early-phase studies. Consequently, it can be difficult to build up a body of evidence for any one clinical indication or health application that would be sufficient to warrant the investment needed for further development or implementation. To maximise the chances of advancing resveratrol, it is important to evaluate the quality, design, and findings of the many trials that have been conducted at both an individual level and when considered collectively. 

The aim of this exercise was to systemically review every clinical trial reported to date that involved the administration of a defined dose of resveratrol in a purified form (i.e., not as part of a mixture) across all possible indications. We aimed to summarise the progress made and current state of knowledge, whilst identifying gaps and opportunities for future clinical research, to help advance the use of resveratrol in situations where efficacy can be demonstrated. 

## 2. Methods

### 2.1. Database Screening and Inclusion/Exclusion Criteria

This systematic review was conducted according to PRISMA guidelines. Searches were conducted by a librarian across Medline, Embase, the Cochrane Database of Systematic Reviews, the Cochrane Central Register of Controlled Trials (CENTRAL), and clinicaltrials.gov, from database inception to 24 February 2023. Relevant reference lists were also searched to ensure complete capture. No limits were applied for date, publication type, or language, but records were limited to human studies. The full search strategies are available in the Appendix A. All articles were imported into the Covidence web-based systematic review software (https://www.covidence.org/, accessed on 30 November 2023), which allows independent dual screening of titles, abstracts, full-text articles, and data extractions. At all stages, any conflicts in screening were resolved by discussion between the lead authors (KB and LH) until the final extracted data set was established. This review and protocol have not been registered on PROSPERO. The PRISMA diagram is shown in Figure 1.

The inclusion criteria were a clinical trial or study on humans; which could be any phase, randomised or non-randomised; studies must involve the administration of pure RSV; different formulations are acceptable, an RSV dose must be specified; and the full publication must be available in the English language. The exclusion criteria were the administration of RSV-containing extract or mixture, including red wine enriched with RSV and use of RSV in combination with another agent, unless there was a comparator group with pure RSV alone; ex vivo, in vitro, and non-human studies; trials examining RSV products for cosmetic purposes; studies in which RSV metabolites were administered as opposed to the parent RSV; review papers; population or observation studies; commentaries; correspondence; and conference abstracts. 

### 2.2. Data Extraction

For data extraction, where articles were not readily available online or through library services, the authors were contacted via email, with follow-up reminders where necessary. Data collection for extraction consisted of the following: sponsorship source; country of origin; clinical setting; type of patients or volunteers; author’s name and institution; year of publication; trial registration identification; study design and groups; whether outcomes were patient-reported; primary outcome/s; whether the study was prospective, retrospective, or a reanalysis of samples/data from a previous study; inclusion/exclusion criteria; adverse event number and type; whether RSV was given with standard of care; total and male/female numbers in each intervention group; mean BMI and mean age of each group; description of the intervention, dosing schedule, total daily dose, and duration of dosing for each group; and for each primary outcome, a change from baseline and between groups. Non-numerical outcomes were also recorded as free text. 

### 2.3. Quality Assessment

Each RCT and associated study was assessed for quality by evaluating the risk of bias in reporting clarity for the following areas: randomisation generation; allocation concealment; blinding of participants; blinding of outcome assessment; incomplete outcome data; selective reporting; and RSV formulation reporting. Each risk of bias was initially judged to be low, high, or unclear, with those reported as unclear by both reviewers subsequently classed as high risk.

## 3. Results 

### 3.1. Number and Geographical Location of RSV Trials

Since 2004, when the first RSV clinical trial was published by Walle et al. [2], an additional 154 individual trials have been reported, with a further 39 published studies describing additional analyses of samples or participant data from these trials (Figure 2A). During this period, a total of 6126 individuals have received RSV under trial conditions at various doses. There has been a general annual trend for increasing numbers of RSV studies, up to a peak in 2018, when 22 unique studies were reported. Since 2011, RCTs have consistently accounted for the majority of RSV studies published during each full year with a smaller proportion of non-RCTs. Interest in RSV is worldwide, as evidenced by the fact trials have been completed in thirty-one countries across five continents (Figure 2B), with the greatest number to date performed in Iran (20), followed by the USA (16), China (15), and Brazil (12). Ten separate trials have been performed in the UK and Australia. The key features of all 194 RSV studies, including details on the participants, dosing regimen, primary objectives, and main outcomes, are summarised in Table 1 (for RCTs) and Appendix A (for non-RCTs). 

### 3.2. Quality Assessment—Risk of Bias

Each published RCT (and associated reanalysis) was assessed for risk of bias against seven specified quality criteria (Figure 3). Almost half (43%) of the studies fulfilled all seven quality requirements and, in total, 86% (118/137) met at least five of the requirements, suggesting that the vast majority of studies had reasonably high reporting standards (Figure 3A). Analysis of the individual responses for each quality indicator revealed that information was mainly lacking on the method of randomisation generation, with 56/137 publications presenting insufficient detail (Figure 3B). Poor reporting of the RSV formulation, in terms of the source or manufacturer, was also apparent in ~23% of studies. 

### 3.3. Clinical Indication and Trial Participants

The first suggestion that RSV might impact human health emanated from the ‘French paradox’, a term that described the 1992 observation by Renaud and de Lorgeril that people in France experienced a relatively low incidence of coronary heart disease despite consuming a diet rich in saturated fats [140]. It was postulated that the moderate wine intake of people in France may explain this finding and RSV, as a constituent of red wine, was proposed to account for the beneficial effects [141]. The validity of the French paradox is still a matter of significant debate; a number of confounders and limitations have been identified within the epidemiology data, but there seems to be evidence of a J-shaped relationship between wine consumption and vascular events and cardiovascular mortality [142]. The underlying mechanisms and protective factors have yet to be defined, but the initial article certainly helped spark interest in RSV and probably contributed to the emphasis that persists today around the cardiovascular and cardiometabolic effects of RSV. 

As of the year 2023, RSV has been investigated for health maintenance, disease prevention, and treatment across a wide range of conditions, which have been broadly grouped in Figure 4 (also see Table 1 and Appendix A). The greatest number of studies (30) have been conducted for the fundamental purpose of characterising pharmacokinetics (PK), distribution, metabolism, and the bioavailability of RSV formulations, whilst often simultaneously evaluating safety as an endpoint. The next most common application is in the management of type 2 diabetes mellitus (T2DM) and glucose control (23 studies), followed by cardiovascular disease (21). Considerable preclinical research has focused on the metabolic effects of RSV and, in addition to the T2DM/glucose studies, this has led to a total of 39 investigations across a spectrum of cardio-metabolic disease, metabolic syndrome, NAFLD, obesity, and, more recently, type 1 diabetes. The general anti-inflammatory and anti-oxidant effects of RSV in humans have also received attention, and its potential for activity in this context has led to trials for conditions, such as ulcerative colitis and arthritis-related diseases. Ten studies have addressed the effects of RSV on cognitive function, and three have explored its ability to modulate biomarkers of Alzheimer’s disease, including one of the most promising larger-scale trials conducted to date in patients with mild to moderate dementia [109,110]. Cancer-related endpoints have been less well studied in humans, particularly considering the wealth of preclinical publications in the area of prevention and treatment across many malignancies, but this may be partly due to the lack of easily accessible and/or surrogate biomarkers of cancer that can be analysed to provide a short-term measure or prediction of efficacy [143]. 

Interestingly, 35% of all RSV studies recruited healthy volunteers; a further 9% specifically involved overweight or obese individuals who were classed as otherwise healthy (Figure 4). Overall, the mean body mass index (BMI) of participant groups recruited to RCTs was higher than individuals in non-RCTs (28.9 versus 26.0), which probably reflects the links between being overweight and risk of developing the types of conditions explored in the RCTs (Figure 5A). Just over half of all RSV studies (56%) involved patients with an underlying disease or pathology (Figure 4). The near-even split between healthy individuals and patients with a diagnosed pathology reflects a focus on health maintenance and prevention, as well as using RSV as a potential treatment. Furthermore, the safety profile of RSV means it has been possible to conduct many of the studies addressing PK, metabolism, and bioavailability in healthy people [144,145,146,147,148]. 

RSV has been investigated in adults of all ages including elderly populations over 70 years, as illustrated by the frequency distribution of mean age per group for individuals that received RSV in each trial (Figure 5B). Furthermore, four separate trials have been conducted in children, assessing the ability of RSV to affect symptoms associated with attention-deficit/hyperactivity disorder [106], autism spectrum disorder [105,149], and fast breathing pneumonia [138].

### 3.4. Duration and Size of Trials

One of the limitations associated with RSV clinical data is that the vast majority of studies conducted have been relatively small, with the median group size for participants who received RSV across all trials being just 22 (Figure 6A). Only a handful of trials (5% of the total) have involved over 100 people taking RSV, which is probably a reflection of the fact RSV is still at a relatively early stage of development for most therapeutic indications. Similarly, most trials have involved a short duration of RSV intervention (median 8 weeks), with the most frequently employed dosing regimen, in 19% of studies, being a single dose, followed by 8 weeks of daily administration (Figure 6B). Six trials involved participants consuming RSV for 12 months [8,11,89,98,108,109] and in a recent small open-label dose–escalation study in 11 patients with muscular dystrophy, RSV was taken for 2 years [150]; this represents the longest reported intervention with RSV within the context of a clinical trial. 

### 3.5. RSV Dose 

RSV has been administered orally at daily doses ranging from 5 to 5000 mg; this represents a 1000-fold difference in daily intake across clinical trials and reflects that the optimum efficacious dose of RSV has not yet been defined for any indication. The most commonly used dose is 500 mg (median is 490 mg), followed by 1 g per day, which were investigated in 32 and 26 trials, respectively. Of all the groups of participants that received RSV in clinical trials, 88% were assigned a dose of 1 g or lower. The trend towards lower doses may have been influenced by relatively early reports that high-dose RSV can modulate the expression/activity of drug metabolising enzymes, together with recommendations not to exceed 1 g per day in healthy populations due to the potential for gastrointestinal side effects, which may affect compliance [143,151]. However, it should be recognised that the choice of the dose is dependent on the balance of risks and benefits; hence, patients undergoing treatment for a particular disease or condition may tolerate a higher degree of toxicity than a healthy person using RSV for health maintenance or prevention. Consequently, an appreciable number of trials have investigated high doses of RSV, with six studies utilising 5 g per day, including two trials in cancer patients involving SRT501, a formulation with significantly enhanced bioavailability compared to standard RSV [114,152]. 

It is notable that only a small number of trials (24 out of a total of 155) included more than one RSV dose to enable direct comparisons of endpoints following the ingestion of different doses. Of these trials, ten were conducted with the primary purpose of establishing ADME/PK or safety, four were classed as pilot studies, and ten were RCTs, meaning less than 10% of all RSV RCTs sought to examine any kind of dose–response relationships. A further eight trials compared different RSV formulations or products, which may also have involved different doses. 

### 3.6. Safety Reporting

Of the 104 individual RCTs conducted to date, 27 listed adverse events occurring in the participants, and another 42 studies specifically stated that no adverse events were reported (Table 1). For the remaining 35 trials, there was no mention of adverse events in the associated publications. There was a similar split for the non-RCTs, where the types of adverse events experienced were detailed in publications from 14 trials; 18 trials stated there were no adverse events, and the remaining 19 failed to report anything on side effects or safety, which may suggest that no obvious or significant AEs occurred within these trials (Appendix A). It is worth noting that the first dose escalation and multiple dosing studies conducted with RSV fall within this non-RCT group; these trials provided the earliest indication of the adverse events associated with oral RSV in volunteers and patients, and the favourable safety profile reported in these trials has been reinforced in subsequent RCTs. 

Of the twenty-seven RCTs that provided detail on the adverse events experienced, four trials involved administering RSV in combination with standard-of-care drugs that have established toxicity profiles, and there were no significant differences in side effects between groups in all of these studies (Table 1). Of the remaining twenty-three trials, eight specifically reported that there were no differences in the incidence of side effects between the the RSV and placebo groups, whilst a further two studies stated that similar numbers of participants reported AEs across the groups. These trials, citing similar adverse event frequencies, used a range of RSV doses, up to a maximum of 3 g daily. 

Across all the trials that reported side effects associated with RSV use, the most common were gastrointestinal in nature, comprising diarrhoea, constipation, nausea, abdominal cramps, vomiting, steatorrhoea, heartburn, reflux, and bloating. In general, studies have shown that RSV is well-tolerated at once-daily doses of ≤1 g, and one of the longest published trials in elderly Alzheimer’s patients has demonstrated an excellent safety profile in doses of up to 1 g taken twice daily for one year [109]. The high acceptability of RSV as a preventive therapy in healthy people is illustrated by the two-year RCT crossover study RESHAW, which examined the effect of 12-month daily RSV supplementation on brain health in post-menopausal women [98]. Of the 146 individuals originally randomised, 86% completed the study, and the most frequent reasons for leaving the trial were lack of time to commit, relocation, and change in pre-existing medical conditions, as opposed to being related to RSV. Furthermore, in the exit survey, 88% of women reported that they would continue with RSV supplementation after the conclusion of the study [98].

### 3.7. Types of Trial Outcomes

Ultimately, to justify the recommendation of RSV for any indication, evidence of clinical efficacy is required; however, the ability to obtain such data is highly dependent on the clinical setting, target population, and whether the intention is prevention, health maintenance, or treatment. Consequently, in many scenarios, studies addressing biomarker changes will be an essential precursor to trials designed to assess efficacy. To provide an overview of how far along the development pipeline RSV has advanced, the outcomes of each published study were classified according to the categories in Figure 7A. There was a tendency for many reports to list multiple primary outcomes with no apparent hierarchy; in this situation, the most robust endpoint has been used to classify the study (i.e., clinical efficacy > clinical biomarkers > exploratory biomarkers).

Thirty out of 194 studies had a primary goal of assessing the safety, pharmacokinetics, bioavailability, or metabolism of RSV, whilst just four addressed drug interactions specifically. The majority (34%) of studies listed the measurement of established markers in routine clinical use, such as insulin resistance, serum oestradiol, liver fat content, and C-reactive protein, as the primary outcome measure (Figure 7A, Table 1 and Appendix A). Another 26% focussed on exploratory markers; these are typically mechanistically driven studies where the marker has not been validated to the point that it is in clinical use for the indication under investigation. Examples include the expression of SIRT in tissues and VEGF protein in muscle, mitochondrial density, serum malondialdehyde as a marker of oxidative stress, and the quantitation of AMPK activation (pAMPK/AMPK ratio) [31,35,74,76]. 

A slightly smaller proportion (22%) of studies had primary outcomes relating to clinical efficacy, such as the healing rate of foot ulcers in people with T2DM, blood pressure changes in pregnant women with pre-eclampsia, the level of endometriosis-related pain, and clinical scores of knee osteoarthritis [6,20,92,117]. 

To provide a broad overview of the potential biological activity of RSV in humans, we evaluated the results from all RCTs, which equates to 137 published studies. RSV was reported as having a significant benefit on the primary outcome in ninety-two studies (67%) (Figure 7B); there was no significant benefit on the primary outcome in forty-two studies (31%), whilst for the remaining three studies, the assessment of benefit was not relevant; for example, in cases where the primary outcome was listed as a safety rather than a pharmacodynamic or clinical endpoint [114]. 

## 4. Discussion

RSV has been used within a trial setting in ~6000 people across the world, over a wide age range, including children and the elderly. Moreover, evidence indicates it is well-tolerated with relatively few side effects recorded and no serious adverse events that can definitively be attributed directly to RSV. There is potential for drug interactions with medications that affect the activity of certain cytochrome P450s (e.g., CYP3A4 and 2E1), but the clinical relevance of these interactions is not yet known, as patients on concomitant medications would have been excluded from many of the trials [153,154]; however, this will need to be considered if the use of RSV becomes more widespread.

RSV pharmacokinetics and metabolism have been well characterised, although there is still a need for the assessment of different dosing schedules and a greater understanding of the potential role of RSV metabolites, both human- and bacteria-derived, in contributing to any clinical efficacy. Recently, it has been proposed that variability in response to RSV may be partly explained by differences in the pattern of metabolites generated. Two metabotypes have been observed, associated with the transformation of RSV by gut microbes, namely lunularin producers and non-producers [155]. Out of 195 healthy volunteers that consumed 150 mg RSV daily for one week, 74% were classed as lunularin producers, and there was a greater prevalence of females in the remaining 26% of participants designated as non-producers. The impact of these metabotypes on the efficacy of RSV remains to be determined, but the findings highlight a need for further studies interrogating the biological activity of individual RSV metabolites, as well as a more comprehensive metabolic profiling of participants in trials to assist in the interpretation of study outcomes. 

Given that RSV has been tested within RCTs alone at doses ranging from 5 mg to 5 g, it is likely that vastly different doses may be required for the management of different therapeutic indications. Few trials have attempted to systematically assess dose–response relationships, beyond those establishing safety or characterising pharmacokinetics/metabolism [146,156], but it is important that a better understanding of this issue is gained with respect to key biological effects and efficacy, particularly considering the demonstration of non-linear dose–response relationships for RSV [157], to enable an evidence-based approach to the selection of the optimal dose for future trials.

### 4.1. Challenges and Knowledge Gaps

Current limitations of the collective clinical data on RSV are the short duration and small size of most trials and the fact that relatively few studies have assessed defined health outcomes that measure whether RSV provides a clinical benefit to patients, using validated surrogate markers for performance, function, morbidity, or mortality from disease. A further weakness is that often the primary endpoint and outcome are not clearly stated or there are multiple listed, with no hierarchy given; this makes it difficult to assess whether a trial has reached its predetermined primary objective. In assessing the risk of bias across RCTs, the quality was generally good overall, but there are areas where the reporting of information was lacking, such as the method of randomisation employed and the precise RSV formulation used and manufacturer, which is needed so that other researchers (and potentially members of the public wishing to use supplements) may access the same products for future trials.

Well-recognised regulatory and financial challenges make it difficult to conduct trials with agents produced and marketed as nutraceutical supplements. There is significantly less economic incentive to fund studies with compounds that often cannot be patented; furthermore, manufacturers do not typically have the level of infrastructure of a pharmaceutical company, financial backing, or appropriate regulatory licences to produce products with evidence-backed health claims. Consequently, the majority of clinical research on RSV is funded by charitable or government organisations and is academic-led, which can limit opportunities for making significant advances. 

The challenges of conducting clinical trials with RSV may be compounded by the sheer breadth of conditions it is being investigated for; this could conceivably dilute the resources available and appetite for funding research on the compound. A strategy to help ameliorate this possibility, which may be particularly relevant in the prevention setting, may be to design trials that assess composite endpoints across multiple conditions with overlapping risk factors, such as obesity-driven cancers, T2DM, and cardiovascular disease, to derive an estimation of wider health benefits. 

As a naturally occurring polyphenol, RSV presents potential pharmaceutical challenges that are common across phytochemicals. Rapid metabolism and subsequent low systemic bioavailability are frequently cited limitations of RSV; however, it is difficult to assess the true impact of its poor bioavailability without better knowledge of the concentrations and doses that are required for clinical efficacy. As alluded to above, these are likely to differ depending on the indication; for example, low oral doses may be sufficient where the aim is to target the gastrointestinal tract since relatively high concentrations of parent RSV have been detected, along with changes in biomarkers associated with oxidative stress, in colorectal tissue of patients that took as little as 5 mg daily [157]. In contrast, in situations where high systemic concentrations are required, doses over 200-fold higher may be required. In these cases, the need to consume large numbers of capsules or tablets is likely to reduce acceptance and uptake by patients/populations whilst also increasing the risk of adverse events, which are dose related. Consequently, the poor bioavailability of RSV may necessitate the use of more advanced oral formulations that delay or prevent metabolic transformation to less or inactive derivatives, with the aim of increasing and prolonging higher systemic concentrations of parent RSV. There is much ongoing research and development in this area, although few formulations have been tested in humans to date [114,158,159,160]. Another consideration that may be more of an issue for aqueous or liquid-based dosage forms is the photosensitivity of RSV, which causes it to isomerise under UV light to the *cis*-isomer, which appears to have limited, if any, biological activity [161]. This will need to be taken into account as more diverse dosage forms are developed, such as gels for topical application and oral dispersions [131,160]. 

### 4.2. Encouraging Findings and Opportunities

Analysis of the types of endpoints utilised in clinical studies and the primary outcomes reported in RCTs (Figure 7) has revealed that the clinical effects and biomarker changes detected in RSV trials are generally in a positive direction, with a benefit reported in two-thirds of cases. Evidence of promising activity was described across a range of indications and since many of the modes of action of RSV are relevant to multiple conditions, results from one clinical setting may have utility in another. As such, despite the small size of the majority of trials conducted to date, recent systematic reviews and meta-analyses are beginning to reveal some reproducible effects of RSV, which may be important across multiple pathologies and conditions. RSV appears to consistently downregulate inflammatory markers, such as C-reactive protein and tumour necrosis factor-α, in a number of disease states, including T2DM, metabolic syndrome, and non-alcoholic fatty liver disease (NAFLD) [162,163,164]. Furthermore, meta-analyses have revealed consistent decreases in plasma glucose and insulin, as well as the regulation of lipid metabolism and improvements in insulin sensitivity with RSV treatment [165,166,167]. Overall, these effects may have benefits in the management of T2DM and other metabolic diseases, and also certain cancers since prolonged inflammatory responses and high circulating glucose play an important role in the promotion of carcinogenesis. In designing future trials, it may be pertinent to note that whilst meta-analyses have shown that RSV does not appear to have a direct effect on weight loss [168], many of the beneficial metabolic outcomes described above are more pronounced in, or even exclusive to, people that are obese or overweight or have a diagnosis of T2DM [167]. Initial early evidence of this differential activity stemmed from a study by Timmers et al. in which RSV at a dose of 150 mg/day was found to mimic the effects of calorie restriction in obese men by lowering energy expenditure and improving their metabolic profile [69], whereas a comparable dose had no effect in non-obese women with normal glucose tolerance [52]. 

As well as the emerging data from meta-analyses, there have also been encouraging findings from several of the larger studies with longer intervention periods, such as the RESHAW trial involving post-menopausal women, which demonstrated that RSV supplementation significantly improved overall cognitive performance, menopausal symptoms, and measures of general well-being [98]. The biomarker changes observed in the one-year intervention trial in patients with mild to moderate dementia due to Alzheimer’s disease also suggest that RSV is worth pursuing in this indication [109]. We are now at the point where a greater number of larger-scale trials with longer durations that assess meaningful health outcomes are needed to advance RSV in areas where it has shown promise. The emphasis of future trials should be on evaluating changes in validated surrogate markers for performance, function, morbidity, and mortality from diseases or conditions rather than changes in biochemical measures in blood or unvalidated surrogate markers. However, such studies can become extremely costly, particularly in situations where regulatory requirements stipulate that RSV is an investigational medicinal product, which must be manufactured to the corresponding regulatory standards. It is, therefore, imperative to ensure that any trials conducted are of high quality and seek to maximise the information gained from every participant, including through the collection of clinical samples for translational research to further mechanistic understanding and ultimately decipher which individuals are likely to benefit from taking RSV. 

Despite the barriers described above, the worldwide interest in RSV shows no signs of abating, with 17 trials currently listed on www.clinicaltrials.gov (accessed on 30 November 2023) as actively recruiting across diverse indications ranging from the prevention of recurrent respiratory tract infections in children to treatment of primary ovarian insufficiency. In addition, in the UK, the cancer-preventive effects of RSV are currently being examined in a phase 2 colorectal polyp prevention trial conducted within the English and Welsh National Health Service Bowel Cancer Screening Programme (BCSP) [169]. The COLO-PREVENT trial will recruit individuals who have attended a screening colonoscopy, at which they are found to have colorectal polyps that are classed as high-risk according to the BCSP standard criteria. Participants (477 in total) will be randomised equally to placebo, low (5 mg), or high (1 g) dose RSV daily for one year, and the primary outcome is the mean number of polyps per participant. The study encompasses a large programme of integrated translational work, including the assessment of circulating and tissue pharmacodynamic biomarkers and repeated metabolic phenotyping and gut microbiome analysis to ascertain whether baseline metabolic status or microbiome profile influences response to RSV and/or whether changes in these parameters may contribute to meditating RSV efficacy. The forward-thinking design parameters of COLO-PREVENT are aimed specifically at bridging the noted gaps in trial quality, size, duration, and use of clinically applicable end-point biomarkers for RSV.

## 5. Conclusions

This review provides a comprehensive summary of all clinical trials conducted on RSV, including the promising evidence of efficacy that has emerged over the past two decades. Despite there being 104 individual RCTs incorporating over 4800 participants on RSV intervention (67% of which have indicated a positive effect when including any reanalyses), there are clear limitations to the studies, both individually and collectively, which have hindered the advancement of RSV towards clinical utility. The existing evidence warrants future clinical studies, preferably ones that are large and have a sufficiently long duration, in patients experiencing multiple conditions, particularly the metabolic diseases that have been described here as being ameliorated by RSV intervention. Such studies should be designed to measure meaningful health outcomes using clinical endpoints or validated surrogate markers of disease. A detailed description of the formulations and randomisation methods employed should be provided to help demonstrate quality and low risk of bias. Additionally, both the potential link between individual patterns of metabolites and treatment outcomes and the evaluation of potential drug interactions deserve close attention.

## Figures and Tables

**Figure 1 ijms-25-00747-f001:**
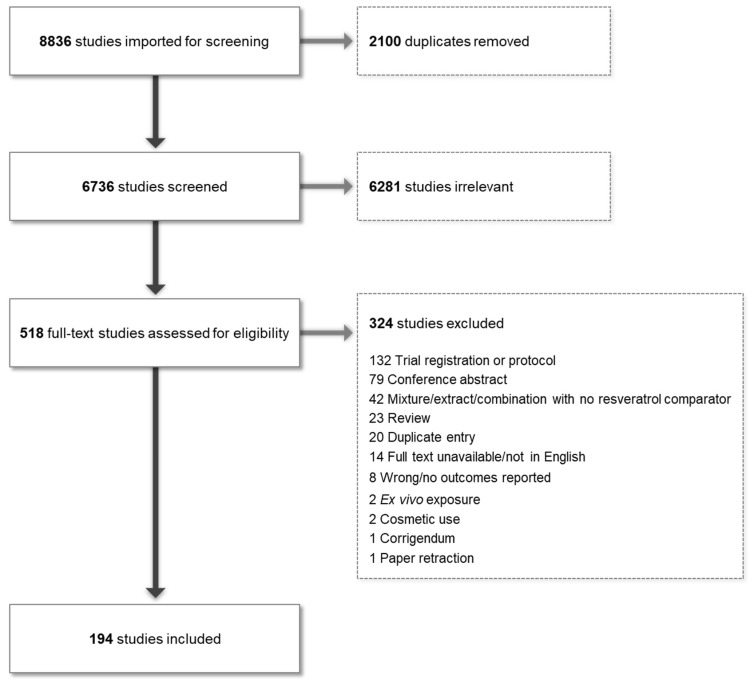
PRISMA diagram showing study flow and reasons for exclusion.

**Figure 2 ijms-25-00747-f002:**
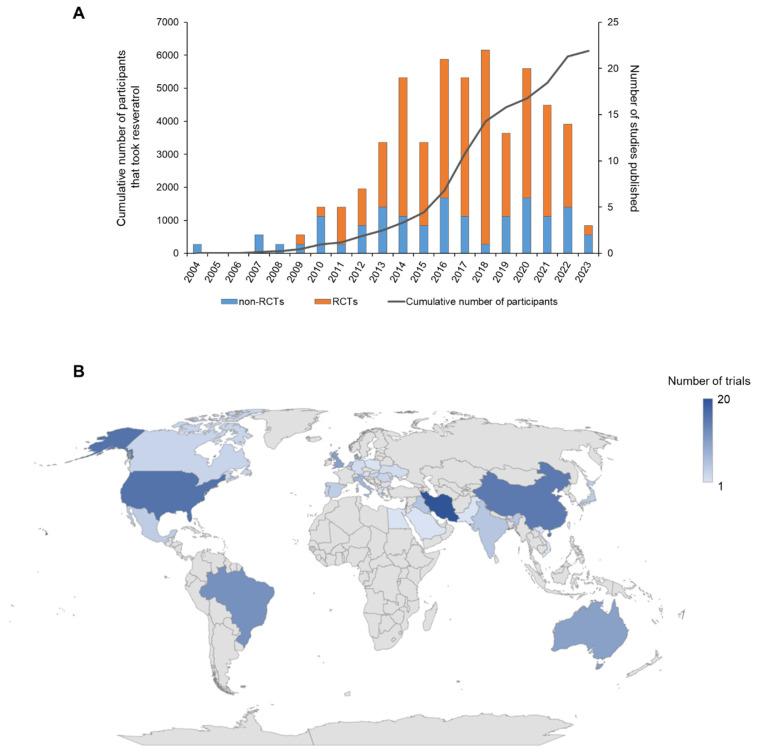
Overview of RSV trials conducted to date. (**A**) The number of clinical studies published on RSV (including reanalyses of participant data or samples) separated into randomised controlled trials (RCTs) and non-RCTs conducted each year since 2004. The graph also shows the cumulative number of participants that took RSV within these studies. (**B**) Geographical spread of all RSV clinical trials, showing the number of unique trials (excluding subsequent studies/reanalyses on the same trial population) performed in each country. Data presented were extracted from all the references cited in Table 1 and Appendix A.

**Figure 3 ijms-25-00747-f003:**
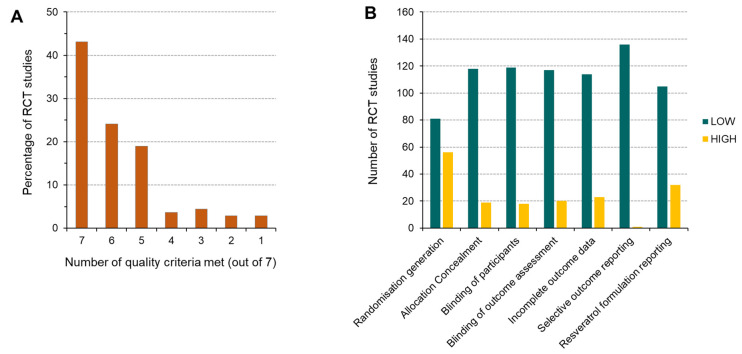
Quality assessment for risk of bias in each randomised controlled trial, including associated reanalysis. Publications were assessed against seven quality criteria, as shown in (**B**). (**A**) The number of quality requirements reached, expressed as the percentage of studies with each score out of 7. (**B**) Breakdown of the risk of bias, defined as high or low, across each of the seven quality criteria for all the RCTs (and reanalyses). The results presented relate to all references cited in Table 1.

**Figure 4 ijms-25-00747-f004:**
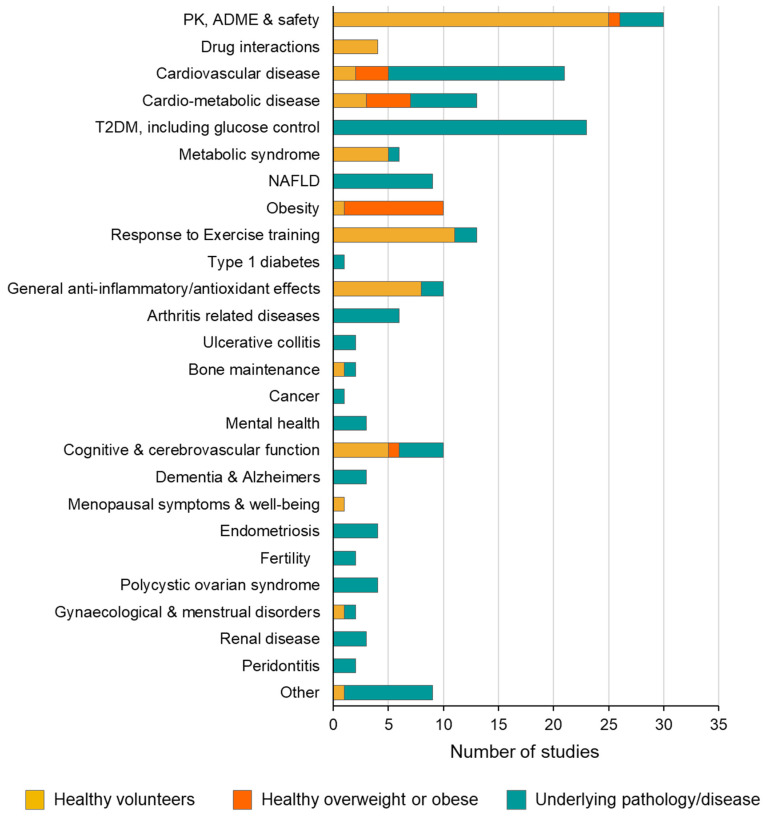
Range of clinical indications investigated in RSV trials and breakdown of the types of participants involved. Each study (including reanalyses) was classified according to its main purpose, such as the treatment or prevention of a particular disease according to the headings shown, or to establish more fundamental characteristics such as pharmacokinetics, ADME, or drug interactions. The studies are further classified according to the participants involved and whether they are healthy volunteers, healthy overweight/obese individuals, or patients with an underlying pathology/disease. Data presented were extracted from all the references cited in Table 1 and Appendix A.

**Figure 5 ijms-25-00747-f005:**
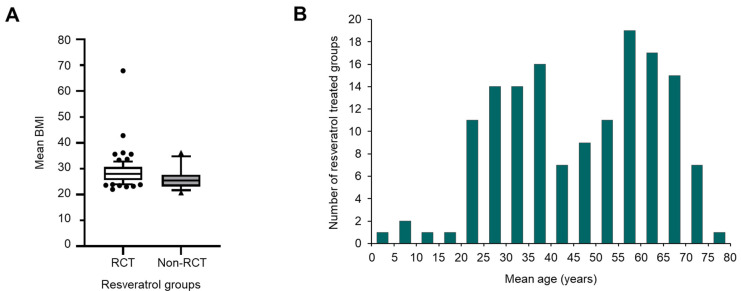
Characteristics of participants in RSV trials. (**A**) Comparison of the mean BMI value of participants taking RSV in RCTs versus non-RCTs. Where there is more than one group receiving RSV, the mean BMI has been calculated for the trial. Values are available from 78 of the 104 RCTs and 19 of the 51 non-RCTs. The graph shows the median value for each group, with the box indicating the 25th to 75th percentiles and the whiskers corresponding to the 10th and 90th percentiles. (**B**) Shows the mean age distribution of RSV-treated participants across all trials where this has been reported. In trials involving multiple groups receiving RSV, the mean age of each group has been included. Mean age data were not available for 39 out of 155 trials. The data presented relate to all references cited in Table 1 and Appendix A.

**Figure 6 ijms-25-00747-f006:**
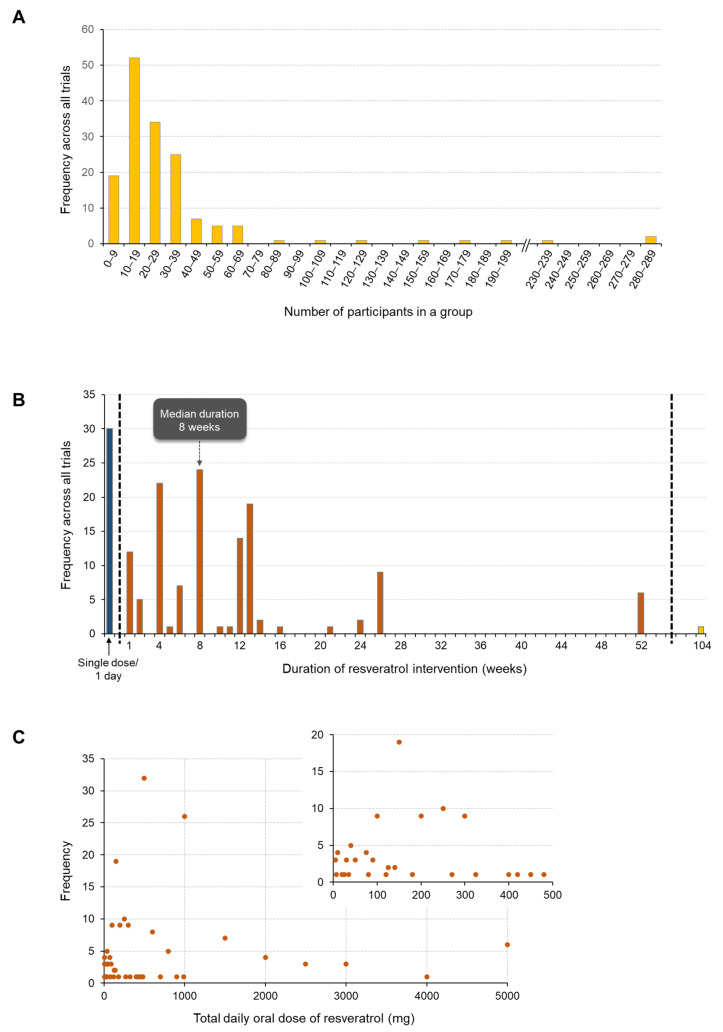
Overview of trial size, duration, and dose used across RSV studies for both RCTs and non-RCTs. (**A**) Frequency distribution of group size across all individual trials. Data represent the number of participants that received RSV; when this equates to multiple groups in a single trial, the average was used. (**B**) Frequency distribution of trial duration in terms of the length of time RSV was taken. The dashed vertical line separates studies involving a single dose given on one occasion from those with repeated dosing lasting more than one day. (**C**) Overview of the different doses used in trials that involved the oral administration of RSV. Results are shown as a frequency distribution for the total daily dose and the inset highlights the range of low doses investigated. Data presented were extracted from all the primary trials listed in Table 1 and Appendix A.

**Figure 7 ijms-25-00747-f007:**
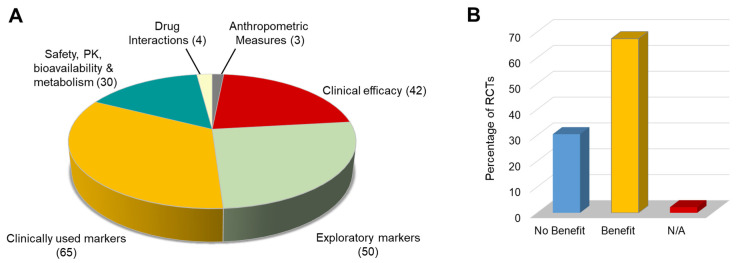
Summary of the types of outcomes evaluated in RSV trials. (**A**) For all studies, including associated reanalyses, the primary outcomes were assigned into one of six categories to provide an overview of the extent of RSV clinical development. Where multiple primary outcomes have been listed in publications with no apparent hierarchy, the most robust endpoint has been used to classify the study where clinical efficacy > clinically used markers > exploratory markers. The figures in parentheses represent the number of studies in each category. (**B**) Proportion of RCTs and associated reanalyses that reported a benefit, or lack of benefit, against the primary outcomes. N/A means that the assessment of benefit was not applicable. Data presented relate to all references cited in Table 1 and Appendix A.

**Table 1 ijms-25-00747-t001:** Summary of all randomised controlled trials (RCTs) identified through the search strategy that involved the administration of resveratrol (RSV) in a purified form. Also listed under each numbered primary RCT publication are any associated studies that used samples and/or data from the original trial, as well as any interim reports that described outcomes. These linked studies are shown with the same background colour (white or grey) and are separated by dashed lines. Trials are grouped according to the clinical indication for which RSV is being investigated in the primary study; in some cases, the associated or follow-up studies may be focussed on a different type of indication but in this table, they have been presented together with the primary trial, for clarity.

	Study (Trial Registration Identification, Where Reported)	Primary Aim or Outcome	Participants	Dose, Schedule, and Formulation	Number of Participants	Adverse Events	Main Findings
**Cardiovascular disease**							
**1**	**Apostolidou 2015** [3]	Changes in blood lipid levels, vitamin E, and total anti-oxidant capacity (TAC)	Healthy volunteers with asymptomatic hypercholesterolemia and normal cholesterol levels	RSV 150 mg or placebo capsule once daily. Crossover design with 30 days of RSV/placebo; 30 days washout then 30 days placebo/RSV	40 in total with 20 in each group	None stated	RSV caused a direct antioxidant effect in normal cholesterol individuals. In asymptomatic hypercholesterolaemic individuals, RSV facilitated an increase in vitamin E
**2**	**Chekalina 2017** [4]	Effects on parameters of central hemodynamics and myocardial ischemia in patients with stable coronary heart disease	Patients with coronary heart disease: stable angina pectoris, FC II, and healthy individuals made up the control group	100 mg RSV daily for 2 months. The pharmaceutical form of RSV is not stated	30 on RSV plus standard therapy; 55 on standard therapy alone	None stated	RSV significantly improved left ventricular systolic function (ejection fraction) and significantly reduced the number of premature atrial and ventricular contractions
**3**	**Chen 2016** [5]	Effect of RSV on improving treatment outcomes of delayed recombinant tissue plasminogen activator (r-tPA) administration. The primary outcome was at least a 4-point improvement from baseline in the NIH stroke scale (NIHSS) score or a complete resolution of symptoms	Brain ischaemic stroke patients	2.5 mg/kg RSV or placebo was given by intravenous bolus injection and then infusion over 60 min	154 on RSV; 158 on placebo	Zero reported	RSV co-administration with r-tPA treatment significantly improved NIHSS scores
**4**	**Ding 2017** [6]	Does RSV provide an effective adjuvant to nifedipine in decreasing blood pressure in severe pre-eclampsia?	Women with pregnancy-induced pre-eclampsia	All patients received nifedipine plus either a 50 mg RSV or placebo capsule every 15 min until blood pressure was ≤150/100 mmHg (maximum of 250 mg RSV)	174 on RSV; 175 on placebo	23 women reported AEs in the RSV group vs. 28 in the control group. No significant differences for maternal or neonatal AEs. RSV AEs were nausea; vomiting; maternal tachycardia; mild headache; dizziness; and chest pain	RSV + nifedipine group needed significantly less time to control blood pressure, delayed the time before a new hypertensive crisis, and resulted in a reduction in the number of doses to control blood pressure compared with the placebo + nifedipine group
**5**	**Djurica 2016** [7]	Primary: reactive hyperaemia index (RHI) with platelet reactivity and plasma nitrate/nitrite	Healthy post-menopausal women	Crossover design; a single dose of 90 mg RSV or an RSV-arginine conjugate in capsule form with a minimum of 1 week washout	Thirty-seven and twenty-five women in two different studies that compared the two formulations both with a crossover design	Not stated	RSV-Arg significantly increased RHI and reduced platelet reactivity compared with RSV
**6**	**Fodor 2018** [8]	Effects on BP, weight status, glucose, and lipid profile	Patients who had a first stroke in the last 12 months	RSV 100 mg or 200 mg once daily for 12 months. A control group (and RSV groups) received standard care only. The pharmaceutical form of RSV is not stated	81 on 100 mg; 55 on 200 mg; and 92 in the control group	Not stated	RSV significantly decreased blood pressure, body mass index, all parameters of lipid profile, and glucose (in non-diabetic patients) compared with the control group
**7**	**Gal 2020** [9]	Effects on left ventricular function and exercise capacity	Patients with systolic heart failure	100 mg RSV or placebo taken in capsule form daily for 3 months	30 on RSV; 30 on placebo	Not stated	RSV significantly improved systolic and diastolic left ventricular function, global longitudinal strain, exercise capacity, ventilation parameters, and quality of life vs. placebo
	***Gal 2020*** [10]	Effects on hemorheological parameters in patients with heart failure with reduced ejection fraction					RSV significantly improved red blood cell aggregation compared to baseline
**8**	**Lixia 2021** [11]	Clinical effects of RSV on the treatment of atherosclerosis	Patients prone to atherosclerosis	100 mg RSV or placebo, plus 20 mg atorvastatin (standard therapy) daily for 12 months. The type of RSV formulation used is not stated	60 in each of the two groups	Not stated	In participants who took RSV plus atorvastatin daily, there was a significant decrease in both systolic and diastolic blood pressure, cholesterol, triglyceride, and low-density lipoprotein (LDL), and a significant increase in aspartate transaminase. There were no significant changes observed in these parameters for participants in the placebo + atorvastatin group after 12 months
**9**	**Magyar 2012** [12]	Endothelium-dependent vasodilatation on systolic and diastolic left ventricular function	Patients with myocardial infarction and angiographically verified coronary artery disease	10 mg RSV or placebo capsule once daily for 3 months	20 on RSV; 20 on placebo	Not stated	RSV significantly improved left ventricular diastolic function and endothelial function as measured by flow-mediated dilatation
**10**	**Marques 2018 (NCT02616822)** [13]	Flow-mediated dilation	Hypertensive patients with endothelial dysfunction	300 mg RSV or placebo in capsule form as a single dose in a crossover study	24	Zero reported	RSV promoted improved endothelial function, especially in women and those with higher LDL-cholesterol
**11**	**McDermott 2017 (NCT02246660)** [14]	Improve six-minute walk performance at 6-month follow-up	Older people with peripheral arterial disease	125 mg, 500 mg RSV, or placebo capsule once daily for 6 months	21 on 125 mg, 23 on 500 mg, and 22 on placebo	14 in the 125 mg group; 27 in the 500 mg group; and 13 in the placebo group. No SAEs were attributable to RSV. AEs were diarrhoea; abdominal pain; and pruritic exanthem	RSV did not improve walking performance
**12**	**Militaru 2013 (ISRCTN0233780)** [15]	Effects on inflammation biomarkers (high-sensitivity C-reactive protein), left ventricular function markers (N-terminal prohormone of brain natriuretic peptide), and lipid markers (total cholesterol, low-density lipoprotein cholesterol, high-density lipoprotein-cholesterol, and triacylglycerols)	Patients with stable angina pectoris	10 mg RSV capsule daily for 60 days. Other groups had 10 mg RSV plus calcium fructoborate, calcium fructoborate alone, or placebo	Twenty-nine patients per group with four study groups	Zero reported	RSV significantly decreased high-sensitivity C-reactive protein in all groups, with brain natriuretic peptide significantly decreased by RSV and by calcium fructoborate alone. This effect was enhanced by their combination
**13**	**van der Made 2015 (NCT01364961)** [16]	Effect on apolipoprotein A-I plasma concentrations	Overweight and slightly obese subjects with low HDL cholesterol concentrations	75 mg RSV or placebo capsule twice daily for 4 weeks, with at least a 4-week washout	45 participants; crossover design	Zero reported	RSV had no effect on apolipoprotein A-1 plasma concentrations
	***van der Made 2017*** *(Reanalysis of samples)* [17]	Effects on endothelial function in the fasting state and postprandial phase					RSV had no effect on flow-mediated dilation in the fasting or postprandial state
**14**	**Wong 2011** [18]	Whether RSV could acutely improve FMD in a dose–dependent manner	Overweight individuals with mildly elevated blood pressure	Each participant consumed three doses of RSV (30, 90, and 270 mg) and placebo in capsule form at weekly intervals	19 participants; crossover design	Not stated	RSV (30, 90, 270 mg doses) significantly increased FMD compared to placebo. There was a significant linear relationship between log_10_ of RSV dose and acute FMD response
**15**	**Wong 2013 (ACTRN12611000060943)** [19]	Whether there is a sustained effect on FMD after 6 weeks of daily RSV	Obese but otherwise healthy men and post-menopausal women	Initially randomised to either 75 mg RSV or a placebo capsule daily for 6 weeks in a crossover trial	28 participants in total	Zero reported	RSV significantly increased FMD
**Type 2 Diabetes Mellitus, including glucose control**							
**16**	**Bashmakov 2014 (ACTRN12610000629033)** [20]	The healing rate of foot ulcers in type 2 diabetics	T2DM patients with diabetic foot syndrome	50 mg RSV or placebo capsule morning and evening for 60 days	14 on RSV; 10 on placebo	None stated	RSV decreased diabetic ulcer size compared with placebo and marginally improved performance in the foot pressure test
**17**	**Bhatt 2012 (CTRI/2011/05/** **00173)** [21]	HbA1c levels	Patients with T2DM	250 mg RSV capsule once daily for 3 months plus standard of care (anti-hypoglycaemic drugs as required). The control group received the standard of care only	30 on RSV; 32 in the control group	None stated	RSV significantly improved mean haemoglobin A1c
	***Bhatt 2013*** [22]	Effects of RSV in patients with T2DM on a range of anthropo-metric and biochemical markers					RSV significantly decreased body weight, systolic blood pressure, cholesterol, triglyceride, urea nitrogen, and total protein
**18**	**Bo 2016 (NCT02244879)** [23]	Effects on CRP	Individuals with T2DM	RSV at a dose of 40 mg or 500 mg once daily, or placebo, given in capsule form for 6 months	65 in each RSV group; 62 on placebo	Zero reported	RSV caused a non-significant dose–dependent reduction in CRP compared with placebo
	***Bo 2017*** [24]	Circulating levels of PTX3					RSV caused a dose–dependent increase in PTX3 and total anti-oxidant status (TAS).
	***Bo 2018*** [25]	Association between changes in SIRT-1 level and variation in H3K56ac value					RSV (500 mg) significantly increased SIRT1. RSV significantly decreased H3K56ac and body fat percentage in the highest SIRT-1 tertial
	***Gambino 2019*** [26]	Impact of rs12778366 (an SNP located in the promoter region of SIRT1) on the response to RSV supplementation in T2DM patients—the primary outcome was SIRT1 protein levels in PBMN cells					SIRT1 decreased in variant C-allele carriers but increased in T-allele homozygotes. Differences between C-allele carriers and T-allele homozygotes were significant in the RSV arm (40 mg)
	***Bo 2018*** [27]	Bone mineral density; circulating concentrations of calcium metabolism biomarkers					RSV (500 mg) caused a significant change in whole-body BMD, whole-body BMC, whole-body T-score, and serum phosphorus compared with placebo between baseline and study end
**19**	**Brasnyo 2011** [28]	Is RSV beneficial for controlling and/or improving insulin resistance?	Male patients with T2DM	5 mg RSV or placebo capsule twice daily for 4 weeks	10 on RSV; 9 on placebo	Zero reported	RSV significantly decreased insulin resistance
**20**	**deLigt 2018 (NCT02129595)** [29]	Improved insulin sensitivity and muscle mitochondrial function	Overweight males at increased risk of developing T2DM	Crossover RCT; participants took 150 mg RSV or placebo in capsule form once daily for 30 or 34 days with a washout of at least 30 days between interventions	13; crossover study	None stated	RSV significantly improved ex vivo muscle mitochondrial function on a fatty acid-derived substrate but did not improve insulin sensitivity
	***Boswijk 2022*** [30]	Reduction in arterial inflammation, assessed by ^18^F-FDG uptake measured by PET	Eight participants out of the original fifteen underwent an additional ^18^F-FDG PET scan and are included in this study				Arterial ^18^F-FDG uptake was non-significantly higher after RSV in comparison to placebo
**21**	**Goh 2014 (NCT01677611)** [31]	SIRT expression in muscle and energy expenditure	Male patients with T2DM	RSV or placebo 500 mg starting dose; the dose was increased by 500 mg/day every 3 days to a maximum dose of 3 g per day in three divided doses if there was no hypoglycaemia. The duration was 12 weeks. Formulation of RSV was not stated but compliance was monitored by ‘pill counting’	5 on RSV; 5 on placebo	Four AEs reported. The proportion of patients with AEs did not significantly differ between groups. AEs were mild elevation of ALT in one patient on RSV; diarrhoea and mild hypoglycaemia, which resolved spontaneously in one patient on RSV; mild cellulitis at biopsy site in one patient per group	RSV significantly increased SIRT1 expression and p-AMPK to AMPK ratio compared with placebo
**22**	**Hoseini 2019 (IRCT20181029041490)** [32]	Insulin resistance	Patients with T2DM and proven 2- and 3-vessel coronary heart disease	500 mg RSV or placebo in capsule form once daily for 4 weeks	30 on RSV; 30 on placebo	Zero reported	RSV significantly reduced fasting glucose, insulin, and insulin resistance. RSV significantly increased insulin sensitivity
**23**	**Khodabandehloo 2018 (RCT2015080223336N2)** [33]	Changes in the percentage of CD14+ CD16+ monocytes; changes to blood glucose	Patients with T2DM	400 mg RSV or placebo capsule twice daily for 8 weeks	25 on RSV; 24 on placebo	Zero reported	RSV caused a significant reduction in fasting blood glucose and blood pressure. There was no change in the percentage of CD14+ CD16+ monocytes
**24**	**Ma 2022 (MDJMU.201905191 × 1)** [34]	Effects of RSV on blood glucose parameters, insulin resistance, nutrient sensing systems, and renal function	Elderly patients with T2DM	500 mg RSV or placebo once daily for 6 months. No information was given on the formulation of RSV used	238 on RSV; 234 on placebo	Although the following RSV-related side effects were reported: diarrhoea (28), constipation (6), muscle cramps/pain (12), fatigue (30), memory loss (4), allergies/upper respiratory infection (4), difficulty swallowing (3), rash (5), headache (4), there were no statistically significant differences in AEs reported from participants on RSV compared to placebo	RSV greatly improved glucose metabolism, insulin tolerance, and insulin metabolism compared to placebo. It significantly decreased levels of glycated haemoglobin/HbA1c, C-reactive protein, IL-6, TNF-α, and IL-1β compared to placebo and improved blood glucose parameters, lipid profile, and renal function
**25**	**Mahjabeen 2022 (SLCTR/2018/019)** [35]	Markers of glucose homeostasis, inflammation, and oxidative stress	Adults (18–70 years) with T2DM of duration ≥5 years, HbA1c 7–12%, and treated with oral hypoglycaemic agents for at least 1 year	One 200 mg capsule RSV or placebo per day for 24 weeks	55 in each group	Zero reported	Significant reductions comparing RSV and placebo groups in HbA1c, fasting insulin, HOMA-IR, hs-CRP, TNF-α, IL-6, and malondialdehyde
**26**	**Movahed 2013 (IRCT201111198129N1)** [36]	Blood glucose lowering effect	Patients with T2DM	500 mg RSV or placebo capsule twice daily for 45 days	34 on RSV; 32 on placebo	Not stated	RSV significantly decreased fasting blood glucose, haemoglobin A1c, insulin, and insulin resistance
**27**	**Sattarinezhad 2019 (NCT02704494)** [37]	Changes in urine albumin/creatinine ratio, estimated glomerular filtration rate (eGFR), and serum creatinine levels	Patients with T2DM and newly confirmed albuminuria	250 mg RSV or placebo capsule twice daily for 90 days	32 on RSV; 32 on placebo	Two AEs consisting of mild dyspepsia	RSV significantly decreased the mean urine albumin/creatinine ratio. eGFR and serum creatinine were unchanged
**28**	**Seyyedebrahimi 2018 (IRCT2015072523336N1)** [38]	Oxidative stress indices in plasma and peripheral blood mononuclear cells	Patients with T2DM	800 mg RSV or placebo in capsule form daily for 2 months	23 on RSV; 25 on placebo	Zero reported	RSV significantly reduced plasma protein carbonyl content and PBMCs O_2_ level and significantly increased plasma total anti-oxidant capacity and total thiol content. The expression of Nrf2 and SOD were significantly increased after RSV consumption
**29**	**Thazhath 2016 (ACTRN12613000717752)** [39]	Effects on GLP-1 secretion, gastric emptying, and glycaemic control in T2DM	Patients with T2DM	500 mg RSV or placebo capsule twice daily for 5 weeks	14; crossover study	Zero reported	RSV did not affect GLP-1 secretion, glycaemic control, gastric emptying, body weight, or energy intake
**30**	**Timmers 2016 (NCT01638780)** [40]	Effect on insulin sensitivity compared with placebo	Men with well-controlled T2DM	75 mg RSV or placebo capsule twice daily for 30 days	17 participants; crossover design	Zero reported	RSV did not improve hepatic or peripheral insulin sensitivity
**Cardiometabolic disease**							
**31**	**Abdollahi 2019 (IRCT20171118037528N1)** [41]	Glycaemic status, lipid profile, and body composition	Individuals with T2DM	500 mg RSV capsule twice daily for 8 weeks and matched placebo	38 on RSV; 38 on placebo	Zero reported	RSV significantly decreased fasting blood sugar and significantly increased high-density lipoprotein compared with placebo. No significant differences in anthropometric measures were observed
	***Tabatabaie 2020*** [42]	Effects on serum levels of ADMA and PON1 activity					RSV significantly decreased serum ADMA and significantly improved PON1 enzyme activity compared with placebo
	***Toupchian 2021;*** [43] *describes the primary objectives for the trial in Abdollahi 2019* [41], *which reports different outcomes*	Effects on the gene expression of PPARα, p16, p53, p21 and serum levels of sCD163/sTWEAK ratio					RSV significantly increased p53 and p21 mRNA expression and significantly decreased serum sCD163/sTWEAK ratio compared with placebo
**32**	**AliSangouni 2022 (IRCT20171118037528N1)** [44]	Effect on hepatic steatosis indices, levels of lipid accumulation product (LAP), and visceral adiposity index (VAI), as well as cardiovascular indices, Casterlli risk index I (CRI-I) and CRI-II, and atherogenic coefficient (AC)	Patients with T2DM	2 × 500 mg capsules RSV or placebo per day for 8 weeks	38 in each group	Zero reported	No significant change in any of the indices measured, LAP, VAI, CRI-I, CRI-II, and AC following RSV, compared to placebo and baseline values
**33**	**deLigt 2020 (NCT02565979)** [45]	Effect on insulin sensitivity	Overweight adults	75 mg RSV or placebo capsule twice daily for 6 months	21 on RSV; 21 on placebo	Zero reported	RSV had no effect on insulin sensitivity
**34**	**Goncalinho 2021 (NCT01668836)** [46]	Effects of RSV compared to energy restriction (ER) diet on vascular reactivity and sympathetic nervous system activity (measured by plasma noradrenaline)	Adults aged between 50 and 65 years with a normal clinical history, physical examination, and resting electrocardiogram	Participants were randomised to 250 mg RSV capsule twice daily or an energy restriction diet (1000 kcal/day) for 30 days	24 in each group	Not stated	RSV does not improve cardiometabolic risk factors, sympathetic activity, or endothelial function. The only effects reported were an increase in apoB and total cholesterol in the RSV group compared to baseline, with no change in plasma noradrenaline
**35**	**Pollack 2017** [47]	Characterise the effects on metabolism, vascular function, and mitochondrial biogenesis	Adults without T2DM	Crossover study design that also included placebo. Initially, 1.5 g RSV in capsule form was taken twice daily by the first nine participants, but the dose was reduced due to gastrointestinal side effects to 1 g twice daily, for 6 weeks	30 participants; crossover study	Three AEs. One SAE of GI symptoms at 3 g/day; one patient requiring hospitalisation. Subsequent dose de-escalation to 2 g/day with no further side effects	RSV improved the fasting reactive hyperaemia index. RSV increased mitochondrial number, and RNA-Seq analysis showed that it significantly perturbs mitochondrial dysfunction and oxidative phosphorylation pathways
**36**	**Poulsen 2013 (NCT01150955)** [48]	Impact of high-dose RSV on energy and substrate metabolism, insulin sensitivity, ectopic fat disposition, 24 h ambulatory blood pressure, and inflammatory and metabolic biomarkers in obese human subjects	Obese healthy male volunteers	500 mg RSV or placebo tablet three times each day for 4 weeks	13 on RSV; 12 on placebo	Four in RSV and four in placebo groups. RSV AEs were flatulence, reflux, and rash.	RSV had no significant effect on insulin sensitivity, endogenous glucose production, turnover and oxidation rates of glucose, blood pressure, resting energy expenditure, oxidation rates of lipid, ectopic or visceral fat content, or inflammatory and metabolic parameters
	***Clasen 2014*** [49]	Effects of RSV on growth hormone signalling in skeletal muscle and adipose tissue					RSV had no significant effect on growth hormone-induced STAT5b phosphorylation or the transcription level of CISH, SOCS2, or IGF-1 in muscle or fat
**37**	**Simental-Mendia 2019** [50]	Effects on lipid profile	Healthy adults with a new diagnosis of dyslipaemia	100 mg RSV or placebo capsule once daily for 2 months	35 on RSV; 36 on placebo	Zero reported	RSV significantly decreased total cholesterol and triacylglycerol
**38**	**Williams 2014** [51]	RSV effects of AMPK/SIRT1 axis in human skeletal muscle and adipose tissue	Overweight participants with less than one hour per week of structured exercise at enrolment	Participants consumed a meal plus a capsule containing 300 mg RSV or placebo, with an interval of at least 7 days between	Eight participants; crossover design	Not stated	RSV had no effect on nuclear SIRT1 activity, AMPK phosphorylation, ACC, or PKA in either skeletal muscle or adipose tissue
**39**	**Yoshino 2012 (NCT00823381)** [52]	Effect on insulin sensitivity and global gene expression and molecular changes in adipose tissue and skeletal muscle	Caucasian post- menopausal non-obese women	75 mg RSV or placebo capsule once daily for 12 weeks	15 on RSV; 15 on placebo	Zero reported	RSV had no effect on liver, skeletal muscle, or adipose tissue insulin sensitivity. RSV did not affect AMPK, Sirt1, NAMPT, and PGC-1α in either skeletal muscle or adipose tissue
**40**	**Zhou 2023 (NCT04886297)** [53]	Changes in the lipid profile	Individuals with dyslipidaemia	RSV (100, 300, or 600 mg) or placebo in capsule form daily for 8 weeks	41 on 100 mg RSV, 43 on 300 mg, 41 on 600 mg, and 43 on placebo	The placebo and RSV were generally well-tolerated, and no serious adverse reactions were reported. Most participants stopped taking the supplements for personal reasons, while the remaining few stopped due to mild adverse effects, with one person each in the placebo and 100 mg RSV groups reporting stomach ache and one person on 600 mg reporting headache	No significant change to the serum lipid profile with RSV intervention compared to placebo. A significant reduction in serum uric acid at 300 mg and 600 mg RSV compared to placebo. No significant differences between RSV and placebo groups for other metabolic markers, glucose, insulin, or oxidative stress biomarkers
**41**	**Zortea 2016 (NCT 02062190)** [54]	Efficacy of RSV on serum glucose and CVD risk factors	Patients with schizophrenia	200 mg RSV or placebo once daily for 4 weeks. No details were given on the formulation used, but compliance was monitored by ‘pill counting’	Ten on RSV; nine on placebo	Zero reported	RSV did not affect glucose or CVD risk factors
	***Zortea 2016*** [55]	Effects on cognition					RSV did not significantly improve neuropsychology performance measures and psychopathology severity after 1 month
Non-Alcoholic Fatty Liver Disease (**NAFLD**)							
**42**	**Asghari 2018** [56]	Whether RSV can improve oxidative/anti-oxidative status in patients with NAFLD	NAFLD patients	600 mg RSV or placebo given in capsule form daily for 12 weeks	30 on RSV; 30 on placebo	None stated	RSV did not cause significant changes to any of the outcomes
**43**	**Asghari 2018 (RCT201511233664N16)** [57]	Effect on anthropo-metric indices and metabolic factors and comparison with calorie restriction	Patients with NAFLD	600 mg RSV or placebo daily given in capsule form; also compared with a calorie-restricted diet for 12 weeks	30 on RSV; 30 on placebo; 30 on calorie restriction	Zero reported	RSV significantly reduced weight and BMI compared with placebo
**44**	**Chachay 2014 (ACTRN12612001135808)** [58]	Insulin resistance	Overweight or obese diagnosed with NAFLD	1.5 g RSV or placebo in capsule form each morning and evening (total daily dose 3 g) for 8 weeks	10 on RSV 10 on placebo	None stated	RSV did not improve any features of NAFLD but increased hepatic stress
**45**	**Chen 2015 (ChiCTR-TRC-12002378)** [59]	Effect of RSV on insulin resistance, glucose, and lipid metabolism	Patients with NAFLD	300 mg RSV or placebo in capsule form twice daily for 3 months	30 on RSV; 30 on placebo	Zero reported	RSV significantly decreased aspartate aminotransferase, glucose, low-density lipoprotein cholesterol, alanine aminotransferase, total cholesterol, and the homeostasis model assessment insulin resistance index
**46**	**Faghihzadeh 2014 (NCT02030977)** [60]	The primary outcome was ALT, but the focus was on inflammatory biomarkers	Patients with NAFLD	500 mg RSV capsule once daily for 12 weeks. All patients were advised to follow an energy-balanced diet and physical activity recommendations during the trial	25 on RSV; 25 on placebo	Zero reported	RSV significantly decreased alanine aminotransferase, inflammatory cytokines, nuclear factor κB activity, serum cytokeratin-18, and hepatic steatosis grade vs. placebo
	***Faghihzadeh 2015*** [61]	The same primary outcome was reported in the 2014 paper above. Emphasis is on CV risk factors, and BMI, WC, cholesterol, and glucose were also reported					RSV significantly reduced alanine aminotransferase (ALT) and hepatic steatosis compared with placebo. There were no significant changes in blood pressure, insulin resistance markers, and TAG in either group
**47**	**Heeboll 2016 (NCT01464801)** [62]	Assess whether RSV would lower plasma transaminases, liver fat content, and the histological NAFLD activity score (NAS) with superiority to placebo	Patients with transaminasemia and suspected NAFLD	500 mg RSV or placebo tablet three times a day for 6 months	15 on RSV; 13 on placebo	33 AEs with RSV; 14 with placebo. RSV is generally well-tolerated, with the total number of patients reporting AEs similar between the RSV (9/15) and placebo group (6/13). AEs were constipation; diarrhoea; abdominal pain; nausea; heartburn; flatulence; less appetite; more appetite; less fatigue; more fatigue; miscellaneous; Bicytopen fever; (see SAE below) dizziness; and hot flushes. There was one SAE in the RSV arm: febrile leukopenia and thrombocytopenia after 10 days of RSV treatment, which recurred upon repeated exposure to RSV	RSV caused a significant decrease in liver lipid content, but there was no difference from placebo
**48**	**Kantartzis 2018 (NCT01635114)** [63]	Change in liver fat content	Overweight and insulin-resistant subjects	150 mg RSV or placebo in capsule form once daily for 12 weeks	54 on RSV; 54 on placebo	73 (RSV); 43 (placebo). There were no safety issues with RSV. The proportion of patients reporting AEs was similar between the groups. The numbers of subjects having at least one AE were 19/54 for placebo and 24/54 for RSV. AEs for RSV were infections and infestations; immune system disorders; metabolism and nutrition disorders; nervous system disorders; vascular disorders; gastrointestinal disorders; respiratory, thoracic and mediastinal disorders; skin and subcutaneous tissue disorders; musculoskeletal and connective tissue disorders; renal and urinary disorders; reproductive system and breast disorders; and injury, poisoning, and procedural complications	Placebo reduced liver fat content, but RSV did not
**49**	**Poulsen 2018 (NCT01446276)** [64]	Long-term effects of high-dose RSV on basal and insulin-mediated very-low-density lipoprotein triglyceride, palmitate and glucose kinetics, and liver fat content in men with NAFLD	Non-diabetic, upper-body obese men with confirmed non-alcoholic fatty liver disease	500 mg RSV or placebo three times each day for 6 months. No details on the formulation used, but tablets are mentioned	Eight on RSV; eight on placebo	Not stated	RSV had no significant effect on basal or insulin-mediated VLDL-TG secretion or its oxidation or clearance rates, palmitate turnover, glucose turnover, body composition, and liver fat content compared with placebo treatment
**Metabolic syndrome**							
**50**	**Batista-Jorge 2020** [65]	Effects on biochemical and physical parameters in combination with changes to diet and exercise regimen	Obese patients with metabolic syndrome	250 mg RSV or placebo capsule daily in combination with a physical activity programme + diet for 3 months	13 on RSV; 12 on placebo	None stated	RSV improved total cholesterol, high-density lipoprotein cholesterol, very-low-density lipoprotein cholesterol, urea, creatinine, and albumin vs. placebo. Anthropometric parameters were significantly different after 3 months of physical activity for both placebo and RSV
**51**	**Mendez-del Villar 2014** [66]	Effects on metabolic syndrome, insulin secretion, and insulin sensitivity	Patients with metabolic syndrome	500 mg RSV or placebo capsules three times each day for 90 days	12 on RSV; 12 on placebo	Zero reported	RSV significantly decreases weight, BMI, fat mass, WC, AUC of insulin, and total insulin secretion
**Obesity**							
**52**	**Arzola-Paniagua 2016 (National Clinical Trial Registry of Mexico 33300410A0152)** [67]	Body weight loss alone and in combination with orlistat	Obese adults	100 mg RSV capsule three times per day with meals, or orlistat alone, the combination of RSV and orlistat, or placebo. All participants were also on an energy-reduced diet	40 on RSV alone; 40 on orlistat alone; 41 on the combination; 40 on placebo	Related and non-related AEs were observed in the intervention groups, with no differences among the groups. No SAEs were reported. The number of patients withdrawing due to AEs was three in the orlistat + RSV group, six in the RSV group, two in the orlistat group, and four in the placebo group. AEs consisted of abdominal pain; constipation; diarrhoea; nausea; and steatorrhoea	RSV + orlistat caused significant weight loss compared to placebo. RSV + orlistat significantly decreased BMI, waist circumference, fat mass, triglycerides, leptin, and the leptin/adiponectin ratio
**53**	**Mahmood 2018** [68]	Obesity-related nephropathy	Obese female patients	1 g RSV in capsule form twice daily for 8 weeks	24 on RSV plus standard treatment; 22 on standard treatment of orlistat, metformin, and fluoxetine	Not stated	RSV significantly reduced serum and urinary creatinine, microalbuminuria, collagen IV, and alpha glutathione-S-transferase (alpha GST), with the estimated glomerular filtration rate (eGFR) significantly elevated. RSV significantly reduced glutamic oxaloacetic transaminase (SGOT), glutamic pyruvic transaminase (SGPT) total cholesterol (TC), triglycerides (TGs), very-low-density lipoprotein (VLDL-c), and resistin and interleukin-6 (IL-6). RSV significantly increased high-density lipoprotein (HDL-c), serum adiponectin, and superoxide dismutase 1 (SOD1)
**54**	**Timmers 2011 (NCT 00998504)** [69]	Whole-body energy expenditure, substrate utilization, ectopic lipid storage, and mitochondrial function, and lipolysis in adipose tissue and skeletal muscle	Obese but otherwise healthy male volunteers	RSV 150 mg or placebo capsule once daily for 30 days. Crossover design with a 4-week washout	11 participants; crossover design	Zero reported	RSV activated AMPK, increased SIRT1 and PGC-1α protein levels, increased citrate synthase activity, and improved muscle mitochondrial respiration ex vivo. RSV increased intramyocellular lipid levels and decreased intrahepatic lipid content, circulating glucose, triglycerides, alanine-aminotransferase, and inflammation markers
	***Konings 2014*** *(Reanalysis of samples from Timmers 2011)* [70]	Adipose tissue morphology and transcriptional profile					RSV significantly decreased adipocyte size and downregulated Wnt and Notch. RSV upregulated cell cycle regulatory genes lysosomal/phagosomal pathway and transcription factor EB
	***Knop 2013*** *(Reanalysis of samples from Timmers 2011)* [71]	Postprandial incretin hormone and glucagon responses					RSV significantly suppressed postprandial glucagon responses without affecting fasting glucagon levels
	***vanPolanen 2021*** *(Reanalysis of samples from 3 trials: Timmers 2011, 2016, de Ligt 2018, NCT00998504, NCT01638780 and NCT02129595)* [72]	Intramyocellular lipid (IMCL) storage					RSV significantly increased IMCL in predominantly type I muscle fibres in the subsarcolemmal and intermyofibrillar region and in PLIN5-coated lipid droplets
	***deLigt 2021*** *(Reanalysis)* [73]	Impact of RSV on the expression of renin-angiotensin-system components in adipose tissue and skeletal muscle compared to placebo in people with obesity					RSV significantly decreased the gene expression of angiotensin-converting enzyme 2 (~40%) and leptin (~30%) in abdominal subcutaneous adipose tissue compared to placebo but did not alter angiotensinogen, ACE, or angiotensin II type 1 receptor (AT1R) expression in AT or skeletal muscles. It was suggested that the effect on ACE2 might dampen SARS-CoV-2 spread in COVID-19
**Response to exercise training**							
**55**	**Alway 2017** [74]	Mitochondrial density; muscle fatigue resistance; cardiovascular function	Healthy older adults	500 mg RSV or placebo capsule once daily (+exercise) for 12 weeks	19 on RSV; 16 on placebo	None stated	RSV increased mitochondrial density and muscle fatigue resistance compared with placebo. RSV combined with exercise might provide a better approach for reversing sarcopenia than exercise alone
**56**	**Gliemann 2013** [75]	Effect on training-induced improvements in cardiovascular health parameters	Healthy, aged, physically inactive men	RSV 250 mg or placebo tablets each morning for 8 weeks. Both groups undertook a programme of high-intensity exercise training	14 on RSV; 13 on placebo	Not stated, but see Olesen below	Placebo + exercise was more effective in increasing maximal oxygen uptake than RSV + exercise
	***Gliemann 2014*** [76]	To assess whether RSV would potentiate an angiogenic response to exercise training		RSV or placebo 250 mg each morning for 8 weeks.	Same study as 2013 but also incorporates another study where individuals followed a normal sedentary lifestyle—nine on RSV, seven on placebo		RSV supplementation did not increase the capillary-to-fibre ratio or muscle VEGF protein. Muscle TIMP-1 protein levels were lower in the RSV group than the placebo group
	***Olesen 2014*** [77]	Metabolic adaptations and anti-inflammatory effects in skeletal muscle of healthy aged subjects similar to exercise training				No participants reported any significant side effects throughout the intervention (covers both studies)	RSV did not elicit metabolic improvements and impaired exercise training-induced improvements in markers of oxidative stress and inflammation in skeletal muscle
**57**	**Harper 2021 (NCT02523274)** [78]	Safety, feasibility, and potential efficacy of RSV combined with exercise training on indices of physical function and skeletal muscle mitochondrial function		One capsule twice daily for 12 weeks plus exercise. Equivalent to 500 mg RSV once or twice daily, or placebo (participants took a combination of RSV and placebo capsules, as required)	Twenty per group (three groups)	AE frequency and type were similar between groups. There were nineteen AEs related or possibly related to RSV across the two dose groups. AEs were gastrointestinal issues, musculoskeletal, and dizziness	RSV + exercise was safe and feasible for older adults with functional limitations and may improve skeletal muscle mitochondrial function and mobility-related indices of physical function
**58**	**Laupheimer 2014** [79]	Effects on immune response and delayed onset muscle soreness	Male athletes	200 mg RSV or placebo in tablet form three times each day for 7 days prior to running a marathon	Four on RSV; four on placebo	Zero reported	RSV did not cause any significant changes compared with placebo
**59**	**Lokken 2021 (NCT03728777)** [80]	Exercise capacity, judged by the change in heart rate (HR) during constant load cycling and changes in maximum oxidative capacity		500 mg RSV or placebo capsule twice daily for 8 weeks; crossover design	10 participants; crossover trial	Both RSV and placebo treatments were well-tolerated, with all AEs remitting spontaneously. No SAEs were reported. AEs observed for RSV and placebo were mild gastrointestinal symptoms, including diarrhoea	RSV did not cause any significant differences in HR
**60**	**Macedo 2014** [81]	Plasma metabolic response and certain indicators of oxidative stress—comparing response to a fitness test pre/post-intervention (anti-oxidant system and oxidative stress biomarkers)	Military firefighters	100 mg RSV or placebo capsule once daily for 3 months	30 on RSV; 30 on placebo	Not stated	RSV decreased IL-6 and TNF-a post-fitness test. RSV decreased glutathione peroxidase activity in both pre- and post-fitness tests.
**61**	**Nicolau 2022** [82]	Alterations in anthropometric parameters, heart measures, blood analytes, leukogram, and plaquetogram indices	Female participants aged between 60 and 80	300 mg RSV or placebo capsule once daily for 60 days	Participants were randomised into four groups based on their level of activity at baseline: no exercise with placebo (*n* = 15), no exercise with RSV (*n* = 16), exercise (*n* = 17), and exercise with RSV (*n* = 14)	Not stated	RSV was not associated with any significant changes in many of the parameters assessed. Blood pressure was increased after 60 days in sedentary women compared to those who exercised, although values remained within the normal range. Mean corpuscular volume and red blood cell distribution width were significantly improved in the groups receiving RSV and exercise compared to the sedentary group on RSV
**62**	**Scribbans 2014** [83]	(i) Effects of RSV on exercise-mediated increases in aerobic capacity in young healthy men; (ii) to examine if RSV impacts exercise-induced adaption in anaerobic capacity and submaximal substrate utilization; and (iii) to determine if RSV impacts skeletal muscle gene expression or adaptations in oxidative/glycolytic capacity in a fibre-specific manner	Healthy, recreationally active men	150 mg RSV or placebo capsule once daily for 4 weeks	Eight on RSV; eight on placebo	Not stated	RSV had no significant effects on any outcomes compared to placebo
**63**	**Storgaard 2022** [84]	Heart rate and fatty acid oxidation, measured using a stable isotope technique during constant workload exercise	Patients with fatty acid oxidation disorders—genetically verified deficiency in very long-chain acyl-CoA dehydrogenase (VLCAD) or carnitine palmitoyl transferase (CPT) II	500 mg RSV or placebo capsule twice daily for 8 weeks each in a crossover study, with a 4-week washout in between	Eight participants in a crossover design	13 reported in total as follows: RSV treatment: headache (2); stomach ache/diarrhoea (4); pins and needles (1); eczema (1); respiratory infection (1) Placebo: headache (1); loss of appetite/diarrhoea (1); fatigue (1); respiratory infection (1)	Neither heart rate nor fatty acid oxidation differed at the end of exercise after treatment with RSV versus placebo
**General anti-inflammatory/anti-oxidant effects**							
**64**	**BaGen 2018** [85]	Effects on inflammatory and anti-inflammation factors after oral implantology	Patients with oral implants	2 mg/kg/d of RSV or placebo for 4 weeks. No information was given on the formulation	82 on RSV; 80 on placebo	None stated	RSV decreased mRNA and serum levels of IL-1β, IL-17A, tumour necrosis factor-alpha (TNF-α), intercellular adhesion molecule-1 (ICAM-1), and immunoglobulin A1 (IgA1). RSV increased mRNA and serum levels of IL-2, IL-6, and IL-10
**65**	**Bo 2013** [86]	Levels of inflammatory and oxidative mediators	Healthy smoking volunteers	500 mg RSV tablet or placebo each morning after fasting overnight for 30 days. A crossover study with a 30-day washout between periods of RSV/placebo	50; with 25 in each group randomised to RSV or placebo first	Zero reported	RSV significantly reduced C-reactive protein (CRP) and triglycerides and increased total anti-oxidant status.
**66**	**DiPierro 2020** [87]	Energy metabolism and oxidative status in RBC	Healthy volunteers	325 mg RSV once daily for 30 days. RSV was taken orally but the pharmaceutical form is not stated	Three	Zero reported	RSV did not significantly change to ATP/ADP ratio or RBC NAD levels
**67**	**Vicari 2020** [88]	Changes in the NIH-CPSI (prostatic symptom index) total score and IPSS (prostate symptom score) total score		19.8 mg RSV or placebo tablet twice daily for 2 months	32 on RSV; 32 on placebo	Not stated	RSV showed significant symptomatic improvement of all NIH-CPSI and IPSS subscale scores
**68**	**Zhang 2017** [89]	Whether RSV can mitigate the effects of occupational ELF-EMF exposure on several inflammatory biomarkers and biomarkers of oxidative stress	Workers exposed to electromagnetic fields and a control population	500 mg RSV or placebo capsule twice daily for 12 months	Within the two groups, participants were randomised into RSV or placebo. In the exposed population, 314 were allocated RSV and 289 placebo. For the control population, the numbers were 263 and 214, respectively	Not stated	RSV significantly reversed the adverse impacts of ELF-EMF and significantly decreased urinary 8-OHdG and F2-isoprostane levels compared with the reference group
**Arthritis related disease**							
**69**	**Hussain 2018** [90]	Assessment of lipid-haematological profile, liver and kidney functions, and a comprehensive short-term follow-up of the clinical and physical examinations, vital signs, body weight alteration, and occurrence of drug-related adverse events. Efficacy was a secondary objective	Patients with knee osteoarthritis	500 mg RSV or placebo in capsule form once daily for 3 months. All participants also had meloxicam as the standard of care	55 on RSV; 55 on placebo	Zero reported	RSV + meloxicam significantly reduced total cholesterol and triglyceride levels compared with placebo. RSV + meloxicam significantly decreased serum levels of GOT, GPT, and ALP, but this was also observed for placebo + meloxicam
**70**	**Khojah 2018** [91]	Biochemical and clinical markers of RA	Patients with rheumatoid arthritis	1 g RSV in capsule form once daily for 3 months	50 on resveratrol plus standard treatment; 50 on standard treatment alone	Zero reported	The 28-joint count for swelling and tenderness, and the disease activity score assessment for 28 joints were significantly lowered by resveratrol. Serum levels of C-reactive protein, the erythrocyte sedimentation rate, undercarboxylated osteocalcin, matrix metalloproteinase-3, tumour necrosis factor alpha, and interleukin-6 were significantly decreased by resveratrol
**71**	**Marouf 2018** [92]	Clinical scores of knee osteoarthritis	Patients with knee osteoarthritis	500 mg RSV or placebo in capsule form once daily for 90 days	60 on RSV; 40 on placebo. Both groups had meloxicam as the standard treatment	Not stated	RSV + meloxicam improves pain and symptom scores in patients with mild-to-moderate knee osteoarthritis compared with placebo
	***Marouf 2018****, same study as above but with a few extra patients* [93]	Pain severity and biochemical markers of inflammation			60 on RSV as above; 50 on placebo (an extra 10)		RSV caused a significant time-dependent decrease in pain severity
**72**	**Marouf 2021** [94]	Correlation between pro-inflammatory markers and clinical outcomes when RSV is used as an add-on therapy to meloxicam	Patients with painful knee osteoarthritis	500 mg RSV or placebo daily taken in capsule form, in addition to the standard therapy of 15 mg meloxicam for 90 days	50 patients in the RSV group; 32 in the placebo group	Not stated	RSV significantly improves both the Knee Injury and Osteoarthritis Outcome Score (KOOS) and the Western Ontario and McMaster Universities Osteoarthritis Index (WOMAC) at the end of treatment compared with baseline values and those reported in the control group post-intervention. The addition of RSV to meloxicam also significantly reduces TNF-α, IL-1β, and IL-6 serum levels compared with day 0 and the placebo group. However, there was only a weak non-significant correlation between serum biomarkers and clinical outcomes
**Cognitive and cerebrovascular function**							
**73**	**Evans 2017 (ANZCTR12615000291583)** [95]	Cerebrovascular responsive-ness and cognitive performance	Post-menopausal women	RSV or placebo 75 mg capsule twice daily for 14 weeks	39 on RSV; 41 on placebo	Not stated (see below)	RSV increased CVR in response to hypercapnic and cognitive stimuli. Cognitive tasks and overall cognitive performance significantly improved
	***Wong 2017*** [96]	Whether RSV could improve aspects of overall well-being such as pain perception, menopausal symptoms, sleep quality, perceived QoL, and depressive symptoms				Zero reported	RSV caused a significant reduction in pain and improved total well-being vs. placebo. Improvements correlated with improved cerebrovascular function
**74**	**Kennedy 2010** [97]	Modulation of mental function and increased cerebral blood flow	Healthy men	Reports two studies: PK and assessing cognitive effects. In the PK study, participants took a single dose of 250 mg or 500 mg RSV (with placebo for blinding) with ~7 days in between. In the second study, participants took single doses of 250 mg, 500 mg RSV, or placebo on three separate days in a crossover design. The order was randomly allocated, and the study was blind. RSV was taken in capsule form	Nine in the PK study and twenty-four in the second study	Not stated	RSV caused dose–dependent increases in cerebral blood flow. There was no change in cognitive function
**75**	**ThaungZaw 2021 (ACTRN12616000679482p)** [98]	Cognitive performance	Post-menopausal women	75 mg RSV or placebo capsule twice daily for 12 months; crossover design	Numbers described in this final report of the trial are 63 who started on RSV and 62 on placebo, which gives a total of 125 participants that received RSV	19 AEs. Eight AEs for placebo and four for RSV in the first supplementation phase (previously published). In the second phase of the study, two AEs for RSV and five AEs for placebo were reported. RSV AEs were itching, menses, prolapsed bladder, a pre-scheduled left eye operation (first phase), the exacerbation of gastric reflux, and pre-scheduled surgery for a posterior heart valve stent insertion (second phase). AEs were not necessarily related to RSV	RSV significantly improved overall cognitive performance compared to placebo
	***ThaungZaw 2020****; Reports secondary endpoints from completed RESHAW trial* [99]	Effects on aspects of overall well-being, such as pain perception, menopausal symptoms, mood and depressive symptoms, sleep quality, and perceived quality of life					RSV significantly reduced composite pain score and was associated with significant improvements in cerebrovascular responsiveness to hypercapnia, somatic menopausal symptoms, and general well-being
	***ThaungZaw 2020;*** *12-month interim analysis of RESHAW trial* [100]	Cerebrovascular function, cognitive performance, and cardiometabolic parameters			73 initially randomised to RSV; 73 randomised to placebo. Crossover study—this is the interim analysis at 12 months of a 2-year study	12 AEs in total. Four AEs in RSV were itching, menses, prolapsed bladder, and pre-scheduled left eye operation. Four AEs in the placebo group were itching, exacerbation of gastric reflux, constipation, and pre-scheduled breast reduction procedure. Four SAEs in the placebo group were urinary tract infection, bowel blockage, oesophageal tear repair, and pre-scheduled operation on the lumbar spine	RSV significantly improved overall cognitive performance and attenuated decline in CVR to cognitive stimuli, which was associated with a significant reduction in fasting blood glucose
	***Wong 2020;*** *Reanalysis of ThuangZaw 2020 at the 12-month timepoint* [101]	Treatment-induced changes to bone parameters					RSV improved bone density in the lumbar spine and neck of the femur and decreased C-terminal telopeptide type-1 collagen levels (bone resorption marker) compared with placebo. The increase in bone mineral density resulted in an improvement in T-score and a reduction in the 10-year probability of hip fracture risk
**76**	**Wightman 2014 (NCT01331382)** [102]	To ascertain whether piperine is capable of enhancing the bioefficacy of RSV with regard to cerebral blood flow and cognitive performance in healthy adults	Healthy adults	Each participant received a single dose of placebo, RSV (250 mg) alone, and RSV plus 20 mg piperine in capsule form on separate days at least a week apart	23 participants; crossover design	Not stated	RSV+ piperine significantly enhanced cerebral blood flow
**77**	**Wong 2016 (ACTRN126140008916)** [103]	To determine the most efficacious dose of RSV to improve cerebral vasodilator responsiveness (CVR) in T2DM	Dementia-free older adults with non-insulin-dependent T2DM	Each participant took three different doses of RSV (75, 150, and 300 mg) and a placebo, in capsule form, on a single occasion at weekly intervals	38 participants; crossover design	Zero reported	RSV significantly increased CVR in the middle cerebral artery at all doses. A RSV 75 mg dose was efficacious in the peripheral cerebral arteries
	***Wong 2016*** [104]	Effects of RSV on neurovascular coupling capacity (CVR to cognitive stimuli), cognitive performance, and correlations with plasma concentrations					RSV (75 mg) significantly improved neurovascular coupling capacity, which correlated with total plasma RSV levels. RSV (75 and 300 mg) enhanced performance on the multi-tasking test battery
**Mental health and neurodevelopmental conditions**							
**78**	**Hendouei 2019 (IRCT20090117001556N104)** [105]	Mean change in the score for irritability subscale from baseline/screening to week 10	Children with autism spectrum disorder	RSV 250 mg or placebo tablet twice daily for 10 weeks. Both groups also had risperidone twice daily	35 on RSV; 35 on placebo	There was no significant difference in the frequency of AEs between the two groups. There were 48 AEs with RSV + resperidone, with the most frequent side effects being restlessness, constipation, and diarrhoea. The most frequent side effects in the placebo group were restlessness, increased appetite, and constipation	RSV did not cause any difference in primary or secondary outcomes compared with placebo
**79**	**Rafeiy-Torghabeh 2020 (IRCT20090117001556N115)** [106]	Main ADHD symptoms, evaluated using the Teacher and Parent versions of ADHD-RS-IV	Children with attention-deficit/hyperactivity disorder	250 mg RSV or placebo tablet twice daily for 8 weeks	33 on RSV; 30 on placebo. Both groups received methylphenidate as a standard treatment	38 (RSV + methyl-phenidate); 47 (placebo + methyl-phenidate). Headache; insomnia; drowsiness; fatigue; dry mouth; decreased appetite; nausea; vomiting; diarrhoea; abdominal pain. The most common complications in both groups were decreased appetite (RSV: 20%; placebo: 27%) and headache (RSV: 17%; placebo: 23%). There was no statistical significance between groups	RSV + methylphenidate resulted in a significant improvement in the time–treatment interaction on all three subscales of the Parent ADHD-RS
**80**	**Samaei 2020 (IRCT20090117001556N103)** [107]	Differences in positive and negative syndrome scale scores from baseline to week 8 between the treatment groups	Patients with schizophrenia	200 mg RSV or placebo capsule for 8 weeks	26 on RSV; 26 on placebo	43 AEs. All AEs were tolerable with mild to moderate severity. The frequency of AEs did not significantly differ between RSV and placebo groups. AEs were headache (4), constipation (7), diarrhoea (5), fatigue (3), nausea (3), increased appetite (9), abdominal pain (6), and nervousness (6)	RSV improves negative, general psychopathology, and total scores to a greater extent than placebo
**Dementia and Alzheimer’s disease**							
**81**	**Gu 2021 (SRRSH2017-0075A)** [108]	The antagonistic effects of RSV on moderate to mild Alzheimer’s disease	Patients with Alzheimer’s disease	500 mg RSV or placebo once daily for 52 weeks. RSV was taken orally but the type of formulation is not stated	15 on RSV; 15 on placebo	88 AEs reported in total; however, for each type, they were equally distributed between the RSV and placebo groups. For example, six infections and infestations were reported in each of the RSV and placebo groups and six nervous system disorders were also described in each group	In the RSV group, there was no significant change in plasma or CSF levels of Aβ40 over the 52 weeks compared to baseline, whilst significant decreases were observed in patients on placebo. There was a significant reduction in brain volume and levels of matrix metallopeptidase 9 in the CSF of patients on RSV compared to placebo. These findings indicate potential neuroprotective effects of RSV
**82**	**Turner 2015 (NCT01504854)** [109]	(1) Safety and tolerability; (2) effect on plasma and CSF Ab42 and Ab40, CSF tau and phospho-tau 181, and volumetric MRI; and (3) examine pharmacokinetics	Patients with mild to moderate dementia due to Alzheimer’s disease	500 mg RSV or placebo capsule once daily with a dose escalation by 500 mg increments every 13 weeks, ending with 1000 mg twice daily for 52 weeks	64 on RSV; 55 on placebo	A total of 657 AEs (490 mild, 139 moderate, 28 severe) were reported (355 on drug, 302 on placebo). A total of 113 out of 119 (95%) participants reported at least one AE. Thirty-six SAEs were reported (nineteen on drug, seventeen on placebo), including three deaths (one on drug, two on placebo), which were not related to the study drug. There were no differences between groups for SAEs. The most common AEs were nausea and diarrhoea (in 42% of individuals on RSV vs. 33% on placebo)	RSV significantly decreased CSF Ab40 and plasma Ab40 levels at week 52. RSV increased brain volume loss
	***Moussa 2017*** *(Reanalysis of samples)* [110]	Effects on pro- and anti-inflammatory cytokines, chemokines, and metalloproteinases in CSF and plasma			The study analysed a subset of patients (19 per group) from the trial		RSV reduced CSF MMP9 and increased macrophage-derived chemokine (MDC), interleukin (IL)-4, and fibroblast growth factor (FGF)-2 compared to placebo at 52 weeks
**Pharmacokinetics, ADME, and safety**							
**83**	**Almeida 2009** [111]	Safety and PK	Healthy volunteers	25, 50, 100, and 150 mg RSV or placebo given in capsule form 6 times a day for 48 h	Eight on each dose (total on RSV was thirty-two, with eight on placebo)	Mild AEs—fifteen for RSV (nine possibly related) and three for placebo; AEs were headache, myalgia, somnolence, epididymitis, dizziness, and occipital headache	C_max_ was 3.89, 7.39, 23.1, and 63.8 ng/mL for each dose, respectively. T_1/2_ was 1–3 h following single-dose, and 2–5 h for repeated dosing. Repeated administration was well-tolerated. Bioavailability was higher after morning administration
**84**	**Anton 2014 (NCT01126229)** [112]	Safety and metabolic effects	Healthy, overweight, older adults	150 mg, 500 mg RSV or placebo given in capsule form twice daily for 90 days	14 on 300 mg RSV, 13 on 1 g RSV daily, and 12 on placebo	17 AEs at 300 mg/day; 12 at 1000 mg/day; 10 in the placebo group. AEs were diarrhoea (5); constipation (3); muscle cramps/pain (4); fatigue (4); memory loss (4); allergies/URI (3); difficulty swallowing (2); rash (2); headache (3); others (7)	RSV significantly decreased blood glucose levels post-treatment compared with placebo
	***Anton 2018*** [113]	Cognitive outcomes					RSV (1000 mg/day) improved psychomotor speed compared to participants taking 300 mg/day or placebo
**85**	**Howells 2011 (NCT00920803)** [114]	Safety and pharmacokinetics	Patients with colorectal liver metastases	5 g SRT501 microparticle RSV powder or placebo sachet mixed with liquid to make a suspension, which is drunk once daily for ~14 days	Six on RSV; three on placebo	Seventeen on RSV; three on placebo. Primarily GI and mild in grade. AEs were anal pruritus; diarrhoea; nausea; chills; lethargy; peripheral neuropathy; rash; skin irritation; flushing	SRT501 was well-tolerated. Mean plasma RSV levels following a single dose of SRT501 were 1942 ± 1422 ng/mL, exceeding those published for equivalent doses of nonmicronised RSV by 3.6-fold
**Endometriosis**							
**86**	**Kodarahmian 2019** [115]	Endometrial and serum levels of MMP2 and MMP9	Women with endometriosis	400 mg RSV or placebo twice daily for 12–14 weeks. The type of formulation was not stated	17 on RSV; 17 on a placebo	Not stated, but see below	RSV decreased the mRNA and protein of both MMP-2 and -9
	***Khodarahmian 2021*** [116]	VEGF and TNF-alpha 2 expression in the eutopic endometrium				Zero reported	RSV caused a significant decrease in VEGF and TNF-α gene and protein levels compared with placebo
**87**	**Mendes da Silva 2017 (NCT02475564)** [117]	Pain score	Endometriosis patients	40 mg RSV or placebo capsule once daily for 42 days	22 on RSV; 22 on placebo	Not stated	RSV is not superior to placebo for the treatment of pain in endometriosis
**Polycystic ovarian syndrome**							
**88**	**Bahramrezaie 2019 (IRCT2016030126860N)** [118]	Effect of RSV on the expression of the VEGF and HIF1 genes in granulosa cells in the angiogenesis pathway of PCOS patients	Infertile patients with PCOS	800 mg RSV or placebo daily given in capsule form for 40 days	31 on RSV; 31 on a placebo	Zero reported	RSV significantly decreased the expression of VEGF and HIF1 genes in granulosa cells vs. the placebo
**89**	**Banaszewska 2016 (NCT01720459)** [119]	Effects on the endocrine and metabolic function of women with PCOS	Individuals with PCOS	1500 mg RSV daily for 3 months. The formulation is described as micronised RSV or placebo in pill form	17 on RSV; 17 on a placebo	Two AEs for RSV. AEs were transient numbness of hands in two patients	RSV significantly decreased total testosterone, dehydroepiandrosterone sulfate, and fasting insulin. RSV significantly increased the insulin sensitivity index
**90**	**Brenjian 2020 (IRCT2016041827453 N1)** [120]	Effects of RSV treatment on pro-inflammatory and ER stress markers	Patients with PCOS	800 mg RSV or placebo in capsule form once daily for 40 days	22 on RSV; 22 on placebo	None stated	RSV decreased serum levels of IL-6, IL-1β, TNF-α, IL-18, NF-κB, and CRP. RSV significantly increased the gene expression of ATF4 and ATF6 and significantly decreased levels of CHOP, GRP78, and XBP1
**91**	**Mansour 2021 (IRCT2017061917139N2)** [121]	Reduction in testosterone from baseline to 3 months. As additional indicators of androgen excess in PCOS, changes in FAI, SHBG, DHEA, LH, and FS from baseline to 3 months	Women in the age range 18–40 years diagnosed with PCOS	1 g RSV or placebo in capsule form once daily for 3 months	39 on RSV; 39 on placebo	Zero reported	RSV did not cause any significant differences compared to placebo in terms of testosterone, ovarian, and adrenal androgens, sex hormone binding globulin (SHBG) levels, the free androgen index (FAI), glycoinsulinemic metabolism, and lipid profile
**Renal disease**							
**92**	**Lin 2016** [122]	Whether RSV has beneficial effects on ultrafiltration (UF) and angiogenesis	Patients undergoing peritoneal dialysis	150 mg or 450 mg RSV, or placebo in capsule form, once daily for 12 weeks	24 on each dose of RSV; 24 on placebo	Four patients discontinued RSV. AEs were diarrhoea; constipation; muscle cramps/pain; fatigue; headache; memory loss	RSV (both dose groups) caused a significant improvement in mean net UF volume and UF rate compared to placebo. High-dose RSV significantly reduced appearance rates of VEGF, Flk-1, and Ang-2 compared to placebo
**93**	**Saldanha 2016 (NCT02433925)** [123]	Nrf2 and NF-kB expression	Non-dialysed patients with chronic kidney disease	500 mg RSV or placebo capsule once daily for 4 weeks. Crossover design with an 8-week washout	Twenty in a crossover trial (eleven on placebo first, nine on RSV first)	Zero reported	RSV does not significantly change NRF2 or NFkB
	***Alvarenga 2022*** [124]	Plasma levels of uremic toxins, indoxyl sulfate (IS), p-cresyl sulfate (pCS), and indole-3-acetic acid (IAA)					RSV did not reduce the plasma levels of IS, pCS, and IAA in non-dialyzed patients with chronic kidney disease
**Gynaecological and menstrual disorders**							
**94**	**Dzator 2022 (ACTRN12620000180910)** [125]	The hormonal migraine burden index—HMBI (number of days with menstrual migraine per month)	Healthy volunteers aged between 18 to 50 with a regular hormonal cycle length between 21 and 35 days and suffer from migraine	75 mg RSV capsule twice daily or placebo, each for a period of 3 months	31 women, crossover study	Not stated	No significant difference in the HMBI or other outcomes assessing migraine-related disability and quality of life when comparing RSV and placebo treatments
**95**	**Ma 2018** [126]	Remodelling of the scarred uterus and improved pregnancy outcomes	Women with a scarred uterus	10 mg RSV or placebo once daily for 3 months. No information was given on the formulation of RSV used	46 on RSV; 32 on placebo	Not stated	RSV reduced uterus scarring in 87.36% of patients and downregulated the plasma levels of β-human chorionic gonadotropin. RSV promoted remodelling of the scarred uterus, regeneration of the endometrium, and improved pregnancy outcomes
**Bone maintenance**							
**96**	**Ornstrup 2014 (NCT01412645)** [127]	Change in bone alkaline phosphatase (BAP)	Men with metabolic syndrome	75 mg or 500 mg RSV, or placebo in tablet form twice daily for 16 weeks	23 on 75 mg RSV; 25 on 500 mg RSV; 26 on placebo	RSV high group—seven; RSV low group—three; placebo group—four. Gastrointestinal complaints were mild and primarily related to increased frequency and/or softer stool	High-dose RSV significantly increased BAP vs. placebo at all time points
	***Kjaer 2015*** [128]	Circulating androgens and prostate size			Reports lower numbers of patients than original study: 21 per group for both 75 mg and 500 mg RSV and 24 on placebo		High-dose RSV decreased serum androstenedione and dehydroepiandrosterone-sulphate (DHEAS) and significantly decreased dehydroepiandrosterone (DHEA)
	***Kjaer 2017*** [129]	Inflammatory markers in plasma and adipose tissue, glycaemic status, circulating lipid parameters, blood pressure, and body composition			Same participant numbers as Kjaer 2015		High-dose RSV significantly increased total cholesterol, low-density lipoprotein (LDL) cholesterol, and fructosamine
	***Korsholm 2017*** [130]	Metabolomic analysis on blood, urine, adipose tissue, and skeletal muscle tissue in middle-aged men with metabolic syndrome			Just involves the 500 mg RSV and placebo groups		RSV reduced sulfated androgen precursors in blood, adipose tissue, and muscle tissue, and increased these metabolites in urine. Markers of muscle turnover were increased with increased intracellular glycerol and the accumulation of long-chain saturated, monounsaturated, and polyunsaturated (n3 and n6) free fatty acids. RSV affected urinary derivatives of aromatic amino acids, reflecting the composition of the gut microbiota
**Periodontitis**							
**97**	**Golshah 2021 (IRCT20130812014333N91)** [131]	The efficiency of Emulgel containing resveratrol in improving clinical gingival inflammatory status; criteria included bleeding on probing (BOP), the gingival index (GI), the hyperplastic index (HI), and probing pocket depth (PPD)	Individuals aged between 12 and 25 years old who received fixed orthodontic treatment, before which they had no clinical signs of gingivitis or periodontitis	Participants were randomised to topical Emugel containing 2% RSV, a control group, and a placebo group receiving a similar Emugel product without RSV. The gel (5 mL) was massaged on the gums for 30 s every night after brushing teeth for 8 weeks	26 on RSV; 24 in the control group; 23 in the Emugel placebo group	Zero reported	Emulgel-containing resveratrol is effective in improving gingival health in orthodontic patients for 8 weeks and can reduce gingival inflammation. The HI and GI scores were significantly decreased at 4 and 8 weeks after the start of the study in the RSV group
**98**	**Zhang 2022** [132]	Improvements in symptoms of periodontitis, measured by clinical attachment level (CAL), the bleeding index (BI), the oral hygiene index-simplified (OHI-S), and probing pocket depth (PPD)	Individuals with periodontitis	Patients randomised to 125 mg, 250 mg, or 500 mg RSV daily, or placebo, in capsule form, for 8 weeks	Forty participants in each of the four groups	There were no significant differences in the AEs observed between the RSV and placebo-treated patients. The most common AEs were nausea and diarrhoea. In total, there were 64 AEs in the RSV groups, but the frequency was not dose-related. Eighteen AEs were reported in the 500 mg high dose group (nausea six; diarrhoea nine; hypotension one; proteinuria two), twenty-two were reported in patients on 250 mg (nausea eight; diarrhoea ten; hypotension one; proteinuria three) and there were eight AEs in the 125 mg dose group (nausea eight; diarrhoea eleven; hypotension two; proteinuria three). Values are not reported for the placebo group	Symptoms of periodontitis determined by CAL, BI, OHI-S, and PPD were improved by all doses of RSV compared to placebo. RSV also significantly decreased specific systemic and local inflammatory markers compared to placebo
**Ulcerative colitis**							
**99**	**Samsami-Kor 2015 (RCT201209154010N10)** [133]	Inflammatory biomarkers and quality of life	Patients with mild to moderate active ulcerative colitis	500 mg RSV or placebo capsule for 6 weeks	25 on RSV; 24 on placebo	Zero reported	RSV significantly reduced plasma levels of TNF-α, hs-CRP, and the activity of NF-kB in PBMCs. RSV significantly decreased the clinical colitis activity index score and increased the inflammatory bowel disease questionnaire—9 (IBDQ-9) score
	***Samsamikor 2016*** *(same study as above, but with more patients included)* [134]	Whether RSV can improve oxidative/anti-oxidative status in subjects with UC			28 in each group		RSV increased serum SOD and TAC and significantly decreased serum MDA. RSV significantly decreased disease activity and increased the quality of life
**Other**							
**100**	**Beijers 2020 (NCT02245932)** [135]	Effects on skeletal muscle mitochondrial function	Patients with chronic obstructive pulmonary disease	75 mg RSV or placebo capsule twice daily for 4 weeks	11 on RSV; 10 on placebo	Zero reported	RSV did not have any significant effects on mitochondrial function
**101**	**Malaguarnera 2018** [136]	Liver health, health-related quality of life (HRQL) and a reduction in depression and anxiety in patients with MHE	Patients with Minimal Hepatic Encephalopathy (MHE)	18.8 mg RSV or a placebo tablet once daily for 90 days	35 on RSV; 35 on placebo, which also contained N-acetylcysteine	Zero reported	RSV caused a significant decrease in the Back Depression Inventory (BDI) and state-trait anxiety inventory (STAI). RSV significantly improved physical function, body pain, general health, vitality, and social function
**102**	**Martinez 2015** [137]	Sperm number, motility, or reduced abnormalities	Males with infertility	50 mg RSV daily or placebo in capsule form for 75 days	18 on RSV, 18 on placebo, and 18 on another investigational product—a hydrogen sulfide prodrug	Zero reported in the RSV group	RSV did not cause any significant change to any parameters
**103**	**Qiang 2018** [138]	Clinical efficacy in improving treatment outcome of oral amoxicillin against childhood fast breathing pneumonia. The primary outcome was defined as treatment failure up to day 3	Children diagnosed with fast-breathing pneumonia	20–60 mg RSV depending on body weight, twice daily for 3 days. The control group had placebo and all patients had amoxicillin. No information was given on the RSV formulation used	282 on RSV; 282 on placebo, both with amoxicillin	Zero reported	RSV + amoxycillin revealed decreased treatment failure rates vs. placebo + amoxycillin
**104**	**Shi 2017** [139]	Disease activity was determined using the Birmingham Vascular Activity Score (BVAS)	Patients with acute Takayasu arteritis	250 mg RSV or placebo capsule once daily for 3 months	121 on RSV; 121 on placebo	Not stated	RSV significantly decreased BVAS scores after week 6

List of abbreviations used in this Table: 8-OHdG, 8-Hydroxy-2’-deoxyguanosine; ACC, acetyl-CoA carboxylase; ACE, angiotensin-converting enzyme; ADMA, asymmetric dimethylarginine; ADME, absorption, distribution metabolism and excretion; AE, adverse event; ALP, alkaline phosphatase; AMPK, AMP-activated protein kinase; Ang-2, angiopoietin-2; AUC, area under the curve; BMI, body mass index; BP, blood pressure; CISH, cytokine inducible SH_2_ containing protein; CSF, cerebrospinal fluid; CVD, cardiovascular disease; DHEA, dehydroepiandrosterone; ELF-EMF, extremely low-frequency electric and magnetic field; FMD, flow-mediated dilation; GLP-1, glucagon-like peptide 1; GOT, glutamic oxaloacetic transaminase; GPT, glutamic pyruvic transaminase; H3K56ac, histone H3 with an acetylated lysine 56; HbA1C, haemoglobin A1c; HOMA-IR, homeostatic model assessment for insulin resistance; IL, interleukin; MDA, malondialdehyde; MMP, matrix metalloproteinase; MRI, magnetic resonance imaging; NAD, nicotinamide adenine dinucleotide; NAFLD, non-alcoholic fatty liver disease; NAMPT, nicotinamide phosphoribosyltransferase; PBMCs, peripheral blood mononuclear cells; PCOS, polycystic ovary syndrome; PGC-1α, peroxisome proliferator-activated receptor-γ coactivator 1α; PON1, paraoxonase-1; PPARα, peroxisome proliferator-activated receptor α; PTX3, pentraxin-related protein; RA, rheumatoid arthritis; SAE, serious adverse event; SIRT1, sirtuin-1; SNP, single nucleotide polymorphism; SOCS2, suppressor of cytokine signalling 2; SOD, superoxide dismutase; STAT5b, signal transducer and activator of transcription 5b; sTWEAK, soluble tumour necrosis factor-like weak inducer of apoptosis; T2DM, type 2 diabetes mellitus; VEGF, vascular endothelial growth factor; WC, waist circumference.

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
