# Peer review of "Resveratrol for the Management of Human Health: How Far Have We Come? A Systematic Review of Resveratrol Clinical Trials to Highlight Gaps and Opportunities"

_ijms, 2024, doi:10.3390/ijms25020747_

Round 1

Reviewer 1 Report

Comments and Suggestions for Authors

The manuscript has been prepared correctly and describes a very current topic. Nowadays, the use of resveratrol is very wide and such knowledge about clinical trials is very valuable. However, I believe that the following issues should be supplemented to increase the value of the work:
1. Table 1 lacks information in what form RSV was administered. Was it a liquid/tablet/other form?
2. Please indicate what research should be completed to better understand the usefulness of resveratrol

Reviewer 2 Report

Comments and Suggestions for Authors

Introduction

The introduction is well written.

Results and discussion

Please include a section concerning The French paradox attributed to Resveratrol.

This manuscript has many pertinent information but must be better organized, must be more structured. It is difficult to read.

Table 1 must be more structured. Please group your studies as example, metabolic, nafld, etc…

To my point of view, the figures will be included after the text corresponding, as example Figure 2 must move after section 3.1. Also, it will be a good point to include the number of references corresponding to information present in the figures.

A synthesize of the main target of resveratrol mentioned in the different studies could be very interesting, you mentioned only sirtuins.

Discussion

The discussion is well written.

Reviewer 3 Report

Comments and Suggestions for Authors

ijms-2780248

Type of manuscript: Review

Title: Resveratrol for the management of human health: how far have we come? A systematic review of resveratrol clinical trials to highlight gaps and opportunities

Authors: Karen Brown*, Despoina Theofanous, Robert G. Britton, Grandezza Aburido, Coral Pepper, Shanthi Sri Undru 1 and Lynne Howells

[Major Concerns]

1.    Resveratrol: This review paper extensively reviews numerous research papers on the clinical usefulness of resveratrol, which stands out prominently. However, it is deemed necessary to address inherent limitations of resveratrol itself. In my opinion, there are two main limitations. Firstly, as a natural substance, the effective concentration of resveratrol is very high, making its practical use as a pharmaceutical agent unlikely. Secondly, as pointed out in reference 1 that I have added below, there is a structural limitation. Trans-resveratrol undergoes easy photoisomerization, and the resulting cis-resveratrol lacks any physiological activity. While these factors raise no doubts about the efficacy of resveratrol, they strongly argue for its very limited potential as a therapeutic drug. Hence, it would be advisable to further elaborate on the weaknesses of resveratrol as a pharmaceutical agent.

1)      Mechanism of Resveratrol-Induced Programmed Cell Death and New Drug Discovery against Cancer: A Review. Int J Mol Sci. 2022 Nov 8;23(22):13689. doi: 10.3390/ijms232213689.

2)      HS-1793, a resveratrol analogue, downregulates the expression of hypoxia-induced HIF-1 and VEGF and inhibits tumor growth of human breast cancer cells in a nude mouse xenograft model. Int J Oncol. 2017 Aug;51(2):715-723. doi: 10.3892/ijo.2017.4058. Epub 2017 Jun 27.

3)      Resveratrol analogue, HS-1793, induces apoptotic cell death and cell cycle arrest through downregulation of AKT in human colon cancer cells. Oncol Rep. 2017 Jan;37(1):281-288. doi: 10.3892/or.2016.5219. Epub 2016 Nov 7.

2.  Abbreviations: Most journals require that an abbreviation be spelled out at its first occurrence in the text, followed by the abbreviation in parentheses. (Exception: If the abbreviation is on the journal's list of permitted abbreviations, this need not be done.) Thereafter, only the abbreviation may be used. Note also that abbreviations need to be independently defined in the abstract and the main text of the paper. Abbreviations need not be introduced if they are not used again. Abbreviations should always be defined and used within the main text of the paper. Figures or tables, on the other hand, require separate definitions independent of the main text. Examples: resveratrol (RSV) at Line 135, etc.

3.    In cases where abbreviations are used within figures or tables, please list these abbreviations along with their corresponding full names in the figure legends or at the bottom of corresponding tables. If there are two or more abbreviations, arrange them in alphabetical order.

[Minor Concerns]

1.    Line 21: ‘1g/day’ should be written as ‘1 g/day’.

2.    Line 94: Define ID.

3.    Line 121: ‘U.S’ should be written as ‘U.S.A.’.

4.    Line 135: ‘resveratrol (RSV)’ should be abbreviated at the Introduction.

5.    Line 199: Define BMI.

6.    Line 347: ‘type 2 diabetics’ should be written as T2DM.

7.   Lines 498-499: When consecutive reference numbers exceed three, omit the intermediate numbers and represent them in abbreviated form. Examples: 163~204, etc.

Overall, the manuscript can be considered to publication after minor revision as indicated above.

Comments on the Quality of English Language

None

Round 2

Reviewer 2 Report

Comments and Suggestions for Authors

This manuscript has been improved and the answers provided by the authors are satisfactory.